



# Patterns of changing surface climate variability from the Last Glacial Maximum to present in transient model simulations

Elisa Ziegler[1,2], Nils Weitzel[1], Jean-Philippe Baudouin[1], Marie-Luise Kapsch[3], Uwe Mikolajewicz[3], Lauren Gregoire[4], Ruza Ivanovic[4], Paul J. Valdes[5], Christian Wirths[6], and Kira Rehfeld[1,2]

[1]Department of Geosciences, University of Tübingen, Tübingen, Germany
[2]Department of Physics, University of Tübingen, Tübingen, Germany
[3]Max Planck Institute for Meteorology, Hamburg, Germany
[4]School of Earth and Environment, University of Leeds, Leeds, United Kingdom
[5]School of Geographical Sciences, University of Bristol, Bristol, United Kingdom
[6]Physics Institute, University of Bern, Bern, Switzerland

**Correspondence:** Elisa Ziegler (elisa.ziegler@uni-tuebingen.de), Kira Rehfeld (kira.rehfeld@uni-tuebingen.de)

**Abstract.**

As of 2023, global mean temperature has risen by about $1.45 \pm 0.12°$C with respect to the $1850 - 1900$ pre-industrial baseline according to the World Meteorological Organization. This rise constitutes the first period of substantial global warming since the Last Deglaciation, when global temperatures rose over several millennia by about $4.0 - 7.0°$C according to proxy reconstructions. Similar levels of warming could be reached in the coming centuries considering current and possible future emissions. Such warming causes widespread changes in the climate system of which the mean state provides only an incomplete picture. Indeed, climate's variability and the distributions of climate variables change with warming, impacting for example ecosystems and the frequency and intensity of extremes. However, climate variability during transition periods like the Last Deglaciation remains largely unexplored.

Therefore, we investigate changes of climate variability on annual to millennial timescales in fifteen transient climate model simulations of the Last Deglaciation. This ensemble consists of models of varying complexity, from an energy balance model to Earth System Models and includes sensitivity experiments, which differ only in terms of their underlying ice sheet reconstruction, meltwater protocol, or consideration of volcanic forcing. While the ensemble simulates an increase of global mean temperature of $3.0 - 6.6°$C between the Last Glacial Maximum and Holocene, we examine whether common patterns of variability emerge in the ensemble. To this end, we compare the variability of surface climate during the Last Glacial Maximum, Deglaciation and Holocene by estimating and analyzing the distributions and power spectra of surface temperature and precipitation. For analyzing the distribution shapes, we turn to the higher order moments of variance, skewness and kurtosis. These show that the distributions cannot be assumed to be normal, a precondition for commonly used statistical methods. During the LGM and Holocene, they further reveal significant differences as most simulations feature larger variance during the LGM than Holocene, in-line with results from reconstructions.

As a transition period, the Deglaciation stands out as a time of high variance of surface temperature and precipitation, especially on decadal and longer timescales. In general, this dependency on the mean state increases with model complexity,





although there is a large spread between models of similar complexity. Some of that spread can be explained by differences in
ice sheet, meltwater and volcanic forcings, revealing the impact of simulation protocols on simulated variability. The forcings
affect variability not only on their characteristic timescales, rather, we find that they impact variability on all timescales from
annual to millennial. The different forcing protocols further have a stronger imprint on the distributions of temperature than
precipitation. A reanalysis of the LGM exhibits similar global mean variability to most of the ensemble, but spatial patterns
vary. However, whether current paleoclimate data assimilation approaches reconstruct accurate levels of variability is unclear.
As such, uncertainty around the models' abilities to capture climate variability likewise remains, affecting simulations of all
time periods, past, present and future. Decreasing this uncertainty warrants a systematic model-data comparisons of simulated
variability during periods of warming.

## 1    Introduction

Understanding the response of the climate system during extended periods of global warming is of vital importance given
current and projected anthropogenic warming. However, the observational record provides an insufficient data basis due to
its short length of only about 150 years and its sparse spatial coverage during the earlier years (e.g. Morice et al., 2012). To
extend the record further back in time requires the use of natural climate archives and proxy-based reconstructions. Such re-
constructions have many associated uncertainties, as well as limited resolution in time and space. Combining proxy records
from different locations into a global field reconstruction introduces additional uncertainties, such as different interpolation
and calibration procedures, age models and proxy biases (Christiansen and Ljungqvist, 2017; Tingley et al., 2012). Climate
models, on the other hand, simulate three-dimensional fields of a wide variety of variables that describe the climate. As such,
they provide continuous estimates of climate that are limited by model physics and parametrizations, but allow detailed in-
vestigations of the climate system and its changes on long timescales. Since their simulation length and resolution depend
mostly on computational resources, simulation protocols up until the Paleoclimate Modelling Intercomparison Project phase
3 (PMIP3) encompassed mostly equilibrium simulations of past climate states in the form of time slices. Experiments with
time-dependent (transient) forcings were limited to short periods like the past millennium or done with accelerated boundary
conditions. The latest iteration, PMIP4, added more and longer experiments with transient boundary conditions. This allows
for more in-depth explorations of past transitions in the climate's mean state such as the Last Deglaciation, which we examine
here using an ensemble of transient simulations.

As the transition from the Last Glacial Maximum (LGM, 23 – 19 kyr before present[1]) to the current warm period of the
Holocene (10.65 kyr BP – present-day), the Last Deglaciation was a period of substantial global climate change. Global
mean surface temperature (GMST) increased by about $4 - 7\,°C$ according to proxy-based reconstructions and climate model
simulations (Fig. 1a, Gulev et al., 2021; Osman et al., 2021; Annan et al., 2022; Tierney et al., 2020; Shakun and Carlson,
2010). The spread among recent estimates is similar, with some leaning towards the higher end with a warming of $7.0 \pm 1.0\,°C$
suggested by Osman et al. (2021) and $6.1\,°C$ (5.7, 6.5) by Tierney et al. (2020) and others towards the lower end as Annan et al.

---

[1]Here, before present (BP) refers to the year 1950, Common Era (CE). AP denotes the opposite, after present.





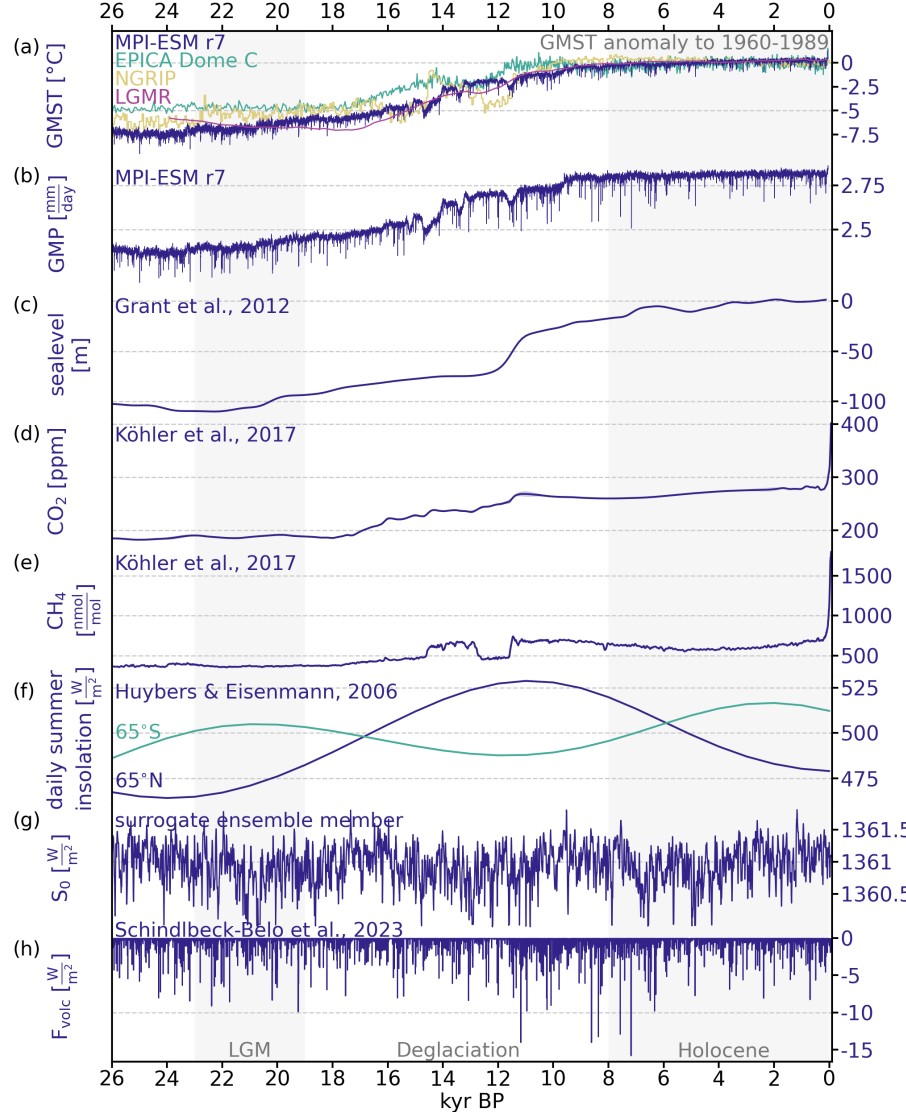

**Figure 1.** Climate responses and external forcing during the past 26k years: (a) global mean temperature anomaly (w.r.t. 1960-1989) as simulated by MPI-ESM, captured in ice cores from Antarctica (EPICA Dome C, Jouzel et al., 2007) and Greenland (NGRIP, Andersen et al., 2004) and reconstructed in the LGM reanalysis (Osman et al., 2021), (b) global mean precipitation as simulated by the Earth System Model MPI-ESM, (c) sea level change (Grant et al., 2012) (d) atmospheric $CO_2$ (Köhler et al., 2017) and (e) $CH_4$ levels (Köhler et al., 2017), (f) daily insolation at 65°N and 65°S at the summer solstice (Huybers and Eisenman, 2006), (g) solar constant from one ensemble member generated as surrogate data based on Steinhilber et al. (2009) following Ellerhoff and Rehfeld (2021) (comparison with Steinhilber et al. (2009) in Fig. S2 in the Supplement), and (h) volcanic forcing TephraSynthIce (Schindlbeck-Belo et al., 2023).



(2022)'s estimated $4.5 \pm 0.9\,°C$. In simulations, a rise in global mean precipitation (GMP) accompanies this warming (Fig. 1b). During the same period, global mean sea level rose by approximately $120\,m$ as the ice sheets in both hemispheres, but especially the Fennoscandian and Laurentide ice sheets, shrunk (Fig. 1c, Lambeck et al., 2014; Grant et al., 2012). However, there are significant uncertainties associated with ice sheet reconstructions, especially with respect to ice sheet extent and elevation (Stokes et al., 2015; Abe-Ouchi et al., 2015; Ivanovic et al., 2016). In turn, timing and magnitude of meltwater

events, which crucially impact the deglacial climate evolution (Snoll et al., 2024), remain uncertain.

Increasing levels of atmospheric carbon dioxide ($CO_2$) contributed to and drove this change (Shakun et al., 2012) as they rose from about $193.2\,ppm$ at the onset of the Last Deglaciation to approximately $271.2\,ppm$ at its end (Gulev et al., 2021). During the Holocene, $CO_2$ levels roughly stabilized until the industrial revolution (Fig. 1d). Similarly, atmospheric methane almost doubled from LGM to Holocene (Fig. 1e, Köhler et al., 2017). Changes in latitudinal and seasonal insolation distribution

favored this rise in atmospheric greenhouse gases and warming (Fig. 1f).

However, considering only the described mean changes is insufficient to capture the full breadth of climate change then and now. Instead, it is necessary to study the climate's variability, too, as reflected in the fluctuations around the mean[2] and in higher order statistics in space and time (Katz and Brown, 1992). These fluctuations determine the actual climate conditions at any point in time and space and are the focus of our study. They affect the various modes of variability (Rehfeld et al., 2020)

and the occurrence and frequency of extremes (Simolo and Corti, 2022; Ionita et al., 2021; Schär et al., 2004; Loikith et al., 2018; Ruff and Neelin, 2012; Laepple et al., 2023).

Climate variability acts across timescales, from intra-annual (i.e. heat waves) and inter-annual (multi-year droughts, ENSO) to millennial scales (D-O events) and beyond, as well as across different spatial scales (Franzke et al., 2020; Laepple et al., 2023). The dependence of variability on timescale suggests that it can be treated as a persistent stochastic process with imposed

trends and quasi-periodic oscillatory modes (Nilsen et al., 2016). The absence of a characteristic timescale in a stochastic process leads to scaling behavior with similar statistical properties across scales (Mandelbrot and van Ness, 1968; Ellerhoff and Rehfeld, 2021). This scaling can, for example, connect slow and fast components of the process in question (Rypdal et al., 2018). It has been found to differ between glacial and interglacial climates, showing that memory effects of the climate system depend on the background state (Nilsen et al., 2016; Roe and Steig, 2004).

Proxy-based reconstructions suggest that global mean temperature variance was about four times higher during the LGM than the Holocene, possibly due to changes in the equator-to-pole temperature gradient (Rehfeld et al., 2018). This implies a dependence of variability on mean climate. The extent of this state-dependency varies regionally, e.g. Rehfeld et al. (2018) find that it is generally larger in the Northern Hemisphere mid- and high latitudes than in the Southern Hemisphere. Models are only partially able to match this LGM to Holocene change in temperature variability. They do agree with reconstructions on

decreasing global temperature variability (Rehfeld et al., 2020) and mean local variability (Ellerhoff et al., 2022) with warming, especially towards higher latitudes (Ellerhoff et al., 2022), but with some exceptions in the tropics (Rehfeld et al., 2020).

---

[2]We here include only changes outside the mean in our use of variability. This is in contrast to IPCC (2021) which includes any deviation from a given equilibrium state, including the change of the mean with time.



Globally, there is mostly agreement between the variance at interannual to centennial timescale in models and reconstructions (Laepple et al., 2023; Zhu et al., 2019). On regional and local scales, however, models simulate less variability than reconstructions, especially on multi-decadal timescales and longer(Laepple et al., 2023; Ellerhoff et al., 2022; Rehfeld et al., 2018). Including natural forcing, that is forcing from solar and volcanic activity, in simulations reduces this difference, but cannot close it (Ellerhoff et al., 2022). Opposite to temperature, precipitation variability increases with warming, with some regional exceptions (Rehfeld et al., 2020). This precipitation variability can be linked to mean precipitation changes, as dry regions generally have lower and wet regions higher variability (Rehfeld et al., 2020).

The influence of natural forcing demonstrates that significant variability arises in response to external forcings and boundary conditions. Volcanism, in particular, has been identified as a prominent driver of changes in temperature, precipitation and modes of atmospheric dynamics (Timmreck, 2012; Iles and Hegerl, 2015; Liu et al., 2016; Zanchettin et al., 2015). Its strongest effects manifest on annual timescales (Lovejoy and Varotsos, 2016), as has been found for the past millennium and Common Era (Schurer et al., 2014; Lovejoy and Varotsos, 2016). It further contributed substantially to subdecadal (Le et al., 2016), decadal (Hegerl et al., 2003) and multidecadal (Schurer et al., 2013) variance. During Glacials, strong volcanic eruptions are even suggested as a driver of millennial variability (Baldini et al., 2015). There is conflicting evidence with respect to the dependence of the impacts of volcanic forcing on the background state: In equilibrium simulations of the LGM and the pre-industrial period (PI), Ellerhoff et al. (2022) found no state-dependency on the global scale for surface temperature variability and only slight differences for precipitation. Bethke et al. (2017), on the other hand, found enhanced variability in future projections on annual to decadal timescales.

Throughout glacial cycles, the cryosphere plays a crucial role for the climate and its variability. This includes sea ice dynamics, as well as changes in ice sheets and associated meltwater releases. Ice sheets and meltwater releases are still commonly simulated as external forcings (Ivanovic et al., 2016). However, reconstructions of ice sheet extent and elevation, as well as associated meltwater pulses entail significant uncertainties (Stokes et al., 2015; Abe-Ouchi et al., 2015; Ivanovic et al., 2016; Izumi et al., 2023). For both the LGM and Last Deglaciation, simulated climate has been shown to be very sensitive to ice sheet reconstructions (Izumi et al., 2023; Kapsch et al., 2022; Bakker et al., 2020; Ullman et al., 2014). Further, meltwater release as a consequence of melting ice sheets affects ocean circulation and thus deglacial climate as a whole (Kapsch et al., 2022). Consequently, the uncertainties in meltwater scenarios and models' varying sensitivities to freshwater crucially affect the simulation of deglacial climate (Snoll et al., 2024).

For sea ice, a decreasing extent has been shown to reduce seasonal to interannual standard deviation of temperature, likely due to polar amplification and the sea ice-albedo feedback (Screen, 2014; Huntingford et al., 2013; Screen and Simmonds, 2010; Bathiany et al., 2018). As a response to shrinking sea ice, this feedback reduces the meridional temperature gradient, which has been linked to decreased variability. Collow et al. (2019) demonstrate a decrease in extreme temperatures, both in frequency and magnitude, with decreasing sea ice extent. Loss of sea ice further leads to an increase in scaling (Rehfeld et al., 2020). In addition, Ellerhoff et al. (2022) found that sea ice dynamics are a significant component of local variability on decadal and longer timescales.



Analyses of variability largely focus on variance, especially in paleoclimate studies, as mean and variance suffice to describe a normal (Gaussian) distribution in full, making variance a useful metric in many contexts. For annual to decadal temperature data, assuming normally-distributed data often is a good approximation after removing periodic variations like the diurnal or seasonal cycle. However, on shorter timescales, this assumption can break down locally and regionally, where many climate
variables are non-normal (Tamarin-Brodsky et al., 2022; Garfinkel and Harnik, 2017; Perron and Sura, 2013; Simolo and Corti, 2022). Such cases necessitate more detailed analyses of the shape of distributions, which higher order moments facilitate.

The higher order moments of skewness and kurtosis facilitate an examination of the asymmetry and heaviness of a distribution's tails, respectively. They have been shown to be pronounced for many atmospheric variables such as geopotential height, vorticity, wind fields and specific humidity (Perron and Sura, 2013) on top of temperature (Tamarin-Brodsky et al., 2022; Ruff
and Neelin, 2012; Skelton et al., 2020; Volodin and Yurova, 2013) and precipitation (He et al., 2013). All else being equal, an increase in variance already increases the probability of extremes, whereas a decrease would counteract it. However, this can be complicated by additional changes in skewness and kurtosis (McKinnon et al., 2016), which generally reveal enhancements or reductions of extremes (Ruff and Neelin, 2012; Simolo and Corti, 2022).

The shape of the tails determines how extremes change with warming, such that, for example under warming, short tails
lead to higher exceedances with respect to fixed hot extreme thresholds than Gaussian tails would (Ruff and Neelin, 2012; Loikith et al., 2018). Additionally, changes in skewness can indicate approaching abrupt shifts. As a system moves towards a tipping point, the weight of the distribution moves towards this point with an increasing long tail away from it, that is skewness increases when approaching a point of abrupt change (Guttal and Jayaprakash, 2008). These kinds of early warning signals have been found in weather station and climate model simulation data (Skelton et al., 2020; He et al., 2013), as well as ecosystems
(Guttal and Jayaprakash, 2008).

Overall, changing dynamics in the Earth System will affect the distributions of a climate variable, potentially resulting in changes in skewness or kurtosis. However, the mechanisms linking the climate system to these statistical measures remain unclear (Simolo and Corti, 2022; Perron and Sura, 2013). For surface or near-surface temperature, asymmetry and long tails are found due to horizontal advection along storm tracks (Garfinkel and Harnik, 2017; Ruff and Neelin, 2012). Land-atmosphere
interactions, in particular changes in soil moisture, are related to significant changes in skewness for near-surface temperature as well (Berg et al., 2014; Douville et al., 2016). Skewness further reflects a marine versus continental influence (McKinnon et al., 2016). Studies of skewness and kurtosis in the literature use data from the 20th century or future projections, often consider only limited timeframes and mostly focus on daily and seasonal data. As a consequence, the role of higher order moments on longer timescales, where normality assumptions might break down, and in past climates is unknown. Whether
they changed between past climate periods, can indicate past abrupt transitions, or could provide a useful metric for intermodel and model-data comparisons remains similarly unclear.

Here, we evaluate how the variability of surface climate changes from the LGM to present, using an ensemble of transient climate model simulations described in Sect. 2.1. We further analyze how model complexity is characterized in Sect. 2.2. Building on this, we investigate how model complexity as well as employed forcing protocol and parametrization affect simulated
variability. To this end, we focus on changes to the distributions of surface temperature and precipitation, including the higher



order moments (Sect. 3.1), and the power spectrum (Sect. 3.2). Based on these changes, we determine how a period of global warming like the Last Deglaciation manifests in the statistical properties of surface climate (Sect. 4). As changes depend on temporal and spatial scales, we consider timescales from annual to centennial and spatial scales from local to global. We then discuss how the variability of surface temperature and precipitation depends on

  – background state (Sect. 5.2),

  – timescale (Sect. 5.2),

  – forcings, in particular ice sheet reconstruction, meltwater forcing protocol and volcanism (Sect. 5.3) and

  – model complexity (Sect. 5.4).

Comparing simulated variability with reconstructions as well as a reanalysis product reveals the impact of forcing protocols on
model-data agreement. Overall, we examine the last global transition in climate to highlight differences between a period of warming in comparison to its preceding and succeeding stable climates.

## 2   Models and data

We draw on an ensemble of fifteen simulations of the Last Deglaciation from climate models of varying complexity (Sect. 2.1). The models range from an energy balance model (EBM) and earth system models of intermediate complexity (EMICs)
to General Circulation (GCMs) and Earth System Models (ESMs), which we evaluate regarding their complexity (Sect. 2.2). Further, we compare the simulations to a multi-proxy reconstruction (Sect. 2.3).

### 2.1   Simulation data

All fifteen simulations are transient and cover at least the Last Deglaciation. They vary regarding simulation setup, applied forcings and model complexity. We separate the simulations into a *main* and *sensitivity set*. Table 1 provides an overview of
the simulations and forcing protocols. The following describes the ensemble in more detail:

  – **MPI-ESM ch4** (Kleinen et al., 2023, 2020)
    **Model**: This *main set* simulation used a setup of MPI-ESM v.1.2 at a coarse resolution called MPI-ESM-CR (Mauritsen et al., 2019; Mikolajewicz et al., 2018) with a methane cycle (Kleinen et al., 2020). Boundary conditions, including ice sheets, bathymetry, topography (Meccia and Mikolajewicz, 2018) from GLAC1-D (Briggs et al., 2014; Tarasov et al.,
2012) and river routing (Riddick et al., 2018) were updated every ten years. It covers 23 kyr BP until present-day.
    **Simulated climate**: This run simulates a LGM to Holocene[3] warming of $4.4°C$ and wetting of $0.27 \frac{mm}{d}$. At its start, it still cools in the global mean in tandem with an increase in the equator-to-pole difference in the Southern Hemisphere and

---

[3]Here, the mean for the LGM and Holocene are computed as defined in Fig. 2, that is, for the LGM we consider 23–19 kyr and for the Holocene 8–0 kyr BP.



in sea ice volume (Fig. 2). It reaches its minimal GMST at around 21 kyr BP. The trend in increasing GMST during the Deglaciation is interrupted by abrupt decreases in GMST thrice: At around 14.5 kyr BP, 13.5 kyr BP and 11.5 kyr BP. In comparison to other simulations, MPI-ESM ch4 simulates the smallest sea ice cover (Fig. 2c). Its global sea ice fraction is largest between 23 and 17 kyr BP and undergoes cycles of abrupt increase and decrease during the Deglaciation.

– **MPI-ESM r1–r7** (Kapsch et al., 2021, 2022)

**Model**: These seven simulations were produced using two more setups of MPI-ESM-CR. They also update boundary conditions every ten years and cover the period from 26 kyr BP to present-day. They use different sets of atmospheric parameters and ice sheet reconstructions – GLAC1-D or ICE-6G_C (in the following ICE6G, Peltier et al., 2015) – and vary by meltwater scenario. The ice sheet reconstructions differ in their original resolution in time, with ICE6G providing updated boundary conditions at 500 yr and GLAC1-D at 100 yr intervals and were here interpolated to 10 yr resolution. The meltwater scenarios follow the options outlined in the deglacial protocol of PMIP4 (Ivanovic et al., 2016): melt-uniform, melt-routed and no-melt. These correspond to meltwater being distributed globally, through river-routing or being removed. Simulation r7 also applies a volcanic forcing as reconstructed by Schindlbeck-Belo et al. (2023) and is part of the *main set*. Runs r1–6 form part of the *sensitivity set*. For two simulations, r3 and r4, only centennial means were available and thus they are considered for the analysis of centennial variability only.

**Simulated climate**: These simulations exhibit the largest warming between LGM and Holocene of the ensemble with a range from 5.3°C (for r5) up to 6.6°C (for r7, Fig. 2a and S1a). The accompanying global mean wetting is also the largest in the ensemble, ranging between $0.30 \frac{mm}{d}$ (r5) and $0.39 \frac{mm}{d}$ (r1 and r7). Runs 1, 6 and 7 further simulate abrupt cooling periods during the Deglaciation with the same timing as in MPI-ESM ch4. These are the simulations that employ the GLAC1-D ice sheet reconstruction and corresponding meltwater forcing. The remaining runs show either continuous or sometimes abrupt warming during those periods. The sea ice cover in these simulations is generally larger than in MPI-ESM ch4 and shows a stronger decrease towards the Holocene.

– **TraCE-21ka** (He, 2010)

**Model**: The TraCE-21ka simulation was performed with CCSM3 (Collins et al., 2006) and stretches from 22 kyr BP to 1990 CE. This *main set* simulation was designed to match proxy data of millennial events such as the Bølling-Allerød and Younger Dryas during the Deglaciation (He, 2010). As such, it applies meltwater forcings in the Northern and Southern Hemisphere at various times throughout the Deglaciation to reproduce proxy records (denoted as melt-routed matched). Ice sheets are updated at intervals of 500 yr based on a modified version of the ICE5G reconstruction (ICE5G*, He 2010; Peltier 2004). As greenhouse gas forcing, TraCE-21ka uses Joos and Spahni (2008) (referred to as J&S in Table 1) with Monnin et al. (2001)'s age model.

**Simulated climate**: Among all simulations TraCE-21ka tends towards the lower end of GMST and GMP change from LGM to Holocene at 4.1°C and $0.20 \frac{mm}{d}$. It shows abrupt warming around the time of the Bølling-Allerød interstadial (about 14.7 – 12.9 kyr BP) with subsequent cooling matching the Younger Dryas (circa 12.9 – 11.7 kyr BP, Fig. 2a). For



most of the time period covered, TraCE-21ka produces the largest sea ice cover, with the exception of the EBM (Fig. 2c). This difference becomes particularly large towards the end of the Deglaciation and remains so throughout the Holocene.

– **HadCM3B r1 & r2** (Snoll et al., 2024)

**Model**: The ensemble contains two simulations from HadCM3B (Valdes et al., 2017) that cover $23\,\mathrm{kyr}$ BP to $2\,\mathrm{kyr}$ AP. These employ two different meltwater protocols, melt-uniform (r1) and melt-routed (r2) from the PMIP4 protocol to match the ICE6G ice sheet history. The simulations prescribe orbit and greenhouse gases (GHG) annually, while ICE6G ice sheet, orography, land-sea mask and bathymetry are updated every $500$ years. HadCM3B r1 is part of the *sensitivity set*, while r2 is included in the *main set*.

**Simulated climate**: The GMST difference between LGM and Holocene is $4.5°$C for the melt-uniform r1 and $4.8°$C for the melt-routed r2. Similarly, wetting of r1 is weaker at $0.26\,\frac{\mathrm{mm}}{\mathrm{d}}$ in comparison to $0.27\,\frac{\mathrm{mm}}{\mathrm{d}}$ for r2. The changes in equator-to-pole gradient are notably small, especially in the Northern Hemisphere (Fig. 2d, e). Sea ice cover shrinks until $14$ kyr BP and remains roughly constant thereafter (Fig. 2c).

– **FAMOUS** (Smith and Gregory, 2012)

**Model**: FAMOUS is a low resolution, slightly simplified version of HadCM3 (Smith et al., 2008). It is sometimes classified as an EMIC (as in Valdes et al. 2017) or as a low-resolution GCM based on the complexity of its atmosphere model. The simulation used here as part of the *main set* was run with an acceleration factor of $10$ for all forcings, allowing it to cover the last $120\,\mathrm{kyr}$ (Smith and Gregory, 2012). The simulation does not consider sea level change, that is, ice sheets are present only where there are no modern ocean grid points. Further, the ICE5G reconstruction and topographic changes (Peltier, 2004) were only applied north of $40°$N (ICE5G**). In particular, the Antarctic ice sheet remains unchanged (Smith and Gregory, 2012).

**Simulated climate**: FAMOUS simulates the smallest global mean change for both surface temperature and precipitation among all simulations at $3.1°$C and $0.15\,\frac{\mathrm{mm}}{\mathrm{d}}$, respectively. Its simulated equator-to-pole temperature differences are among the largest in the ensemble, but they decrease comparatively little from LGM to present-day (Fig. 2d, e).

– **LOVECLIM DG_ns** (Menviel et al., 2011)

**Model**: This version of the EMIC LOVECLIM couples the atmosphere model ECBilt (Opsteegh et al., 1998) to the ocean model CLIO (Campin and Goosse, 1999; Goosse et al., 1999; Goosse and Fichefet, 1999). LOVECLIM DG_ns used ECBilt-CLIO v.3 coupled to a dynamical vegetation model with a terrestrial carbon cycle (Menviel et al., 2011) and is included in the *main set*. It focuses on the Deglaciation, running from $18 - 6.2$ kyr BP. Employed greenhouse forcing is based on reconstructions from EPICA (Lüthi et al., 2008; Monnin et al., 2001; Spahni et al., 2005) mapped onto the EDC3 age scale (Parrenin et al., 2007). Like TraCE-21ka, it includes meltwater pulses in the North Atlantic and Southern Ocean (melt-routed matched) to reproduce millennial scale events in the North Atlantic (McManus et al., 2004) and Greenland (Alley, 2000) during this period (Menviel et al., 2011).

**Simulated climate**: As a result of the employed meltwater pulses, there are a warming and subsequent cooling event visible in the global mean around the times of the Bølling-Allerød and Younger Dryas, respectively (Fig. 2a, b). This





signal is very strong in the Northern Hemisphere, where LOVECLIM DG_ns exhibits a large reduction in equator-to-pole temperature gradient alongside these abrupt changes (Fig. 2d). In the Southern Hemisphere this decrease is more subdued (Fig. 2e). Overall, the simulation shows deglacial warming and wetting comparable to most of the other simulations (Fig. 2a, b).

– **ECBilt-CLIO sim2bl** (Timm and Timmermann, 2007)

**Model**: The second ECBilt simulation included in the *main set* uses the same coupled ocean and atmosphere models and covers the period from $21 - 0$ kyr BP (Timm and Timmermann, 2007). It contains no meltwater forcing. For the ice sheets and land-sea mask of the atmosphere model, it applies ICE4G with the east Siberian ice sheet removed (ICE4G*). For the ocean model, the same ice sheet is used but combined with a constant land-sea mask representing present-day conditions (ICE4G* & PD).

**Simulated climate**: Its simulated mean changes are $3.9°C$ and $0.25\,\frac{mm}{d}$. Like LOVECLIM DG_ns it has deglacial warming and wetting comparable to most of the other simulations (Fig. 2a, b). In magnitude, changes in ECBilt-CLIO sim2bl resemble those in LOVECLIM DG_ns (Fig. 2). Their structure is quite different, though, as ECBilt-CLIO sim2bl variables all change in a step-like manner. The simulated sea ice cover is at the upper end of the ensemble at the beginning of the simulation (similar to TraCE-21ka and MPI-ESM r7) and then, like MPI-ESM r7, reduces drastically towards the

Holocene (Fig. 2c).

– **TransEBM** (Sec. S2.1 in the Supplement)

    **Model**: To represent the linear temperature response of the climate system to external forcing, we juxtapose a simulation from an extended version of the 2D energy balance model TransEBM (Ziegler and Rehfeld, 2021) with the other simulations and include it in the *main set*. Here, it has been extended to include freshwater and zonal volcanic forcing. The

simulation covers the surface temperature evolution of the last 26 kyr, with ICE6G boundary conditions updated every $125$ or $500\,yr$. Sea ice extent was interpolated between the LGM and present-day states given by Zhuang et al. (2017). Meltwater forcing was assimilated based on the database of sea surface temperatures records by Jonkers et al. (2020) (Jonkers assimilated, c.f. Sec. S2.1 in the Supplement). The simulation employs the same volcanic forcing as MPI-ESM r7. Sec. S2.1 in the Supplement describes the simulation in more detail.

**Simulated climate**: TransEBM simulates a GMST difference between LGM and Holocene of $4.1°C$, which is at the lower end of the ensemble and comparable to that of TraCE-21ka. Changes in equator-to-pole difference are similar in magnitude in both hemispheres unlike most other simulations (Fig. 2d, e). Its sea ice cover is the largest and changes the most during the Deglaciation, because EBM models sea ice as a surface type, which covers any given grid cell completely.

To summarize, the *main set* is made up of MPI-ESM ch4 and r7, TraCE-21ka, HadCM3B r2, FAMOUS, LOVECLIM DG_ns, ECBilt-CLIO sim2bl and TransEBM. MPI-ESM r1–r6 and HadCM3B r1 form the *sensitivity set*.




**Table 1.** Forcings applied for the transient simulations of the Last Deglaciation. Further description in Sect. 2.1. Labels of main set simulations are bold.

| Simulation | Forcing | | | | | | |
| --- | --- | --- | --- | --- | --- | --- | --- |
| | Orbital | Solar | GHG | Volcanic | Land-sea mask | Ice sheets | Meltwater |
| **MPI-ESM ch4** | Berger | B&L | Köhler | - | GLAC1-D | GLAC1-D | melt-routed |
| MPI-ESM r1 | Berger | B&L | Köhler | - | GLAC1-D | GLAC1-D | melt-routed |
| MPI-ESM r2 | Berger | B&L | Köhler | - | ICE6G | ICE6G | melt-routed |
| MPI-ESM r3 | Berger | B&L | Köhler | - | ICE6G | ICE6G | no-melt |
| MPI-ESM r4 | Berger | B&L | Köhler | - | ICE6G | ICE6G | melt-uniform |
| MPI-ESM r5 | Berger | B&L | Köhler | - | ICE6G | ICE6G | melt-routed |
| MPI-ESM r6 | Berger | B&L | Köhler | - | GLAC1-D | GLAC1-D | melt-routed |
| **MPI-ESM r7** | Berger | B&L | Köhler | PalVol V.1 | GLAC1-D | GLAC1-D | melt-routed |
| **TraCE-21ka** | Berger | constant | $CO_2$, $N_2O$, $CH_4$; J&S | - | ICE5G* | ICE5G | melt-routed matched |
| HadCM3B r1 | Berger | constant | PMIP4 | - | ICE6G | ICE6G | melt-uniform |
| **HadCM3B r2** | Berger | constant | PMIP4 | - | ICE6G | ICE6G | melt-routed |
| FAMOUS | Berger | constant | $CO_2$, $N_2O$, $CH_4$; EPICA | - | ICE5G** | ICE5G** | - |
| **LOVECLIM DG_ns** | Berger | Berger | $CO_2$; EPICA, other const. | - | ICE4G | ICE4G | melt-routed matched |
| **ECBilt-CLIO sim2bl** | Berger | Berger | $CO_2$, $N_2O$, $CH_4$; Taylor Dome | - | ICE4G* & PD | ICE4G* | - |
| **TransEBM** | Berger | Steinhilber surrogate | Köhler | PalVol V.1 | ICE6G | ICE6G | Jonkers assimilated |



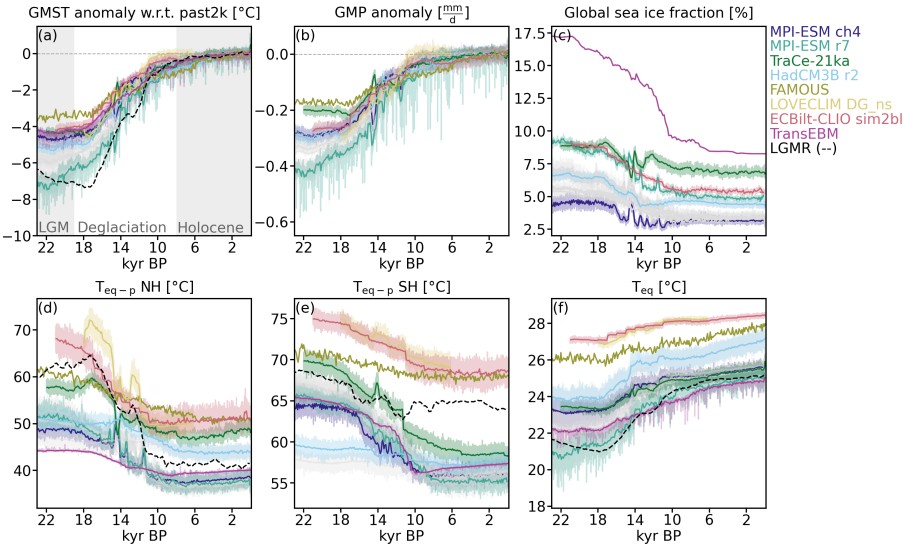

**Figure 2.** Centennial changes in the *main set* from the simulation ensemble from LGM to Holocene with annual data as shading. (a) GMST and (b) GMP anomaly with respect to the past 2 kyr. (c) Global sea ice fraction. Note that the EBM only allows complete coverage of grid cells by one surface type and therefore has the largest sea ice cover. FAMOUS and LOVECLIM DG_ns are missing since their sea ice cover was not readily available. (d, e) Equator-to-pole temperature difference for the Northern and Southern Hemisphere respectively, computed as the difference between polar (70°-90°) and equatorial temperatures (15°S-15°N). The latter are shown in (f). The *sensitivity set* are shown in gray and can be found in Fig. S1 in the Supplement. LGM (23–19), Deglaciation (19–8) and Holocene (8–0 kyr BP) as used in this study are marked in (a).

## 2.2 The model hierarchy

We construct a hierarchy of the models to summarize the outlined differences and thus understand their effects on all aspects of climate variability. The complexity of the models and simulations differs along several axes: resolution in time and space, complexity of the individual components (e.g. atmosphere, ocean, land surface), their coupling and their forcing. Constructing a hierarchy of models or simulations helps summarize those differences and thus understand their effect on any given analysis. The relevant axes of comparison might differ between applications. As a consequence, ranking the same models and simulations might produce a different hierarchy depending on the application. Here, we establish a hierarchy focused on features that affect variability and for which the simulations meaningfully differ.

Based on these considerations, we include eight axes of comparison (Fig. 3a). Sect. S2.2 of the Supplement explores these axes and the classification of the simulations. The resulting hierarchy reveals the various levels of complexity of the different simulations by placing them along each axis. In general, a simulation is considered more complex, the larger the total area it covers. Whenever an axis of the hierarchy does not apply to a simulation, the rank will be at the center of the net, c.f. the lack of dedicated ocean or land hydrology model in TransEBM.





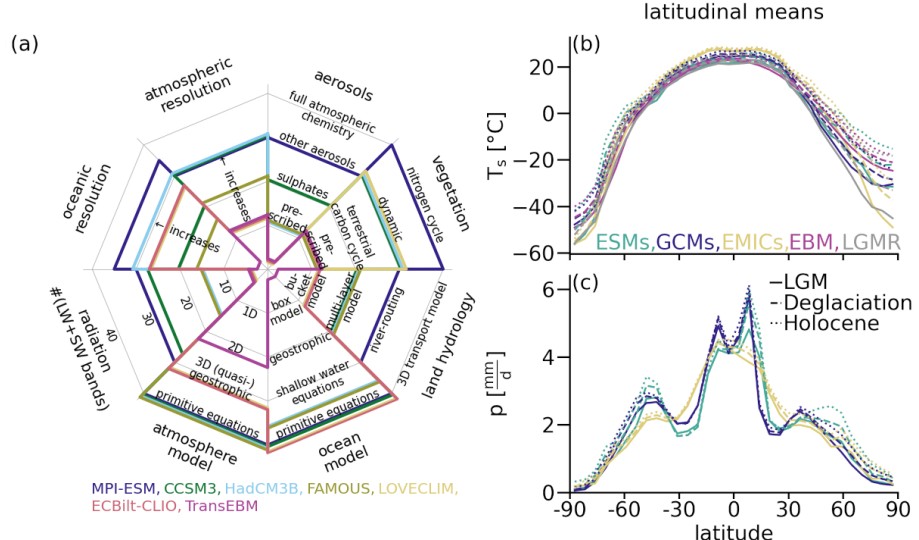

**Figure 3.** (a) Ranking of the models used in this study along eight axes of complexity. The criteria are described in detail in Sect. 2.2 and Table S1 in the Supplement. Altogether they establish a hierarchy of the different models: ESMs (MPI-ESM), GCMs (CCSM3, HadCM3B, FAMOUS), EMICs (LOVECLIM, ECBilt-CLIO) and EBM (TransEBM). (b, c) Latitudinal distributions of mean temperature and precipitation for hierarchy categories based on (a). For temperature the biggest spread between the models and largest overall increases from LGM to Holocene can be found in the polar regions. For precipitation, the simulations and periods vary most in the tropics and the mid-latitude bands.

Based on all the factors summarized in Fig. 3a, we separate all simulations into four groups of complexity for parts of our analysis: ESMs (MPI-ESM), GCMs (TraCE-21ka, HadCM3B1, FAMOUS), EMICs (LOVECLIM, ECBilt-CLIO) and EBM (TransEBM). The categorization follows the overall number of levels reached in the hierarchy. In the end, both applied forcings and complexity of the model components decide the simulation output. Our analysis tries to identify and disentangle the effects of both on simulated variability with the goal of identifying the complexity both necessary and sufficient for long, transient climate simulations. Since increased complexity implies higher computational demand, a trade-off has to be made between complexity and available resources. Knowing the benefits and limitations of added complexity is thus crucial.

### 2.3 Global climate reanalysis data

For quantitative comparison, we draw on a spatio-temporally gridded product, the LGM reanalysis LGMR by Osman et al. (2021), which covers the past 24,000 years. LGMR combines model simulations and proxy reconstructions in an offline data assimilation approach for a proxy-constrained estimate of the full field of surface temperature since the LGM. The resulting dataset has a resolution in time of 200 years, allowing for a comparison of centennial variability to the results of our analysis. The reanalysis relies on model priors from 17 time-slice experiments from iCESM1 (Brady et al., 2019) as well as 539 geo-chemical proxy records of sea surface temperature. Using a Bayesian forward model, proxy values are estimated for given time



steps at every proxy location from the model prior. This produces a forward-modeled proxy value different from the actual

proxy value. To take uncertainties as well as the covariance between proxy location and the climate field into account, this

difference is weighted by the Kalman gain for the update of the model prior temperature field. The resulting reanalysis esti-

mates a global warming of $7.0 \pm 1.0°\mathrm{C}$ from the end of the LGM to PI, as it is contains a LGM state colder than reconstructed

elsewhere (c.f. Annan et al., 2022; Tierney et al., 2020; Shakun and Carlson, 2010). However, unlike other reconstructions,

LGMR provides a gridded reconstruction of the surface temperature field covering the whole time period of interest here, not

just the LGM.

## 3   Methods

Climate can be represented by sets of observations in space and time. The field of a climate variable then refers to usually

gridded spatial representations of that variable (von Storch and Zwiers, 1999). Conversely, a timeseries specifies the sequence of

observations in time (Chatfield, 2016). As such, climate variables can be treated as random variables with associated probability

distributions and timeseries represent realizations of a stochastic process. Here, we analyze the statistical properties of the

timeseries of surface temperature and precipitation in space and time by computing their moments and power spectra.

In order to compare the transient simulations, we first re-grid them to a common T21 resolution, which is the lowest com-

monly used resolution in the ensemble. We further compute decadal and centennial means of the annual data to obtain the

variability on those timescales. Then, we extract the time periods, LGM $(23 - 19\,\mathrm{kyr}\,\mathrm{BP})$, Deglaciation $(19 - 8\,\mathrm{kyr}\,\mathrm{BP})$ and

Holocene $(8 - 0\,\mathrm{kyr}\,\mathrm{BP})$, from all timeseries (c.f. Fig. 2a). Finally, we remove the trend from the timeseries using a Gaussian

filter with a kernel length equivalent to $4000\,\mathrm{yr}$[4], which is the length of the LGM as the shortest time period we investigate.

After detrending, we can assume that the resulting timeseries are (weakly) stationary, a requirement for the estimation of

moments, as well as the autocovariance function and thus the spectrum.

### 3.1   Moments of a probability distribution

The distributions of surface temperature and precipitation cannot be assumed to be normal. Precipitation in particular often has

heavier tails than a normal distribution (Franzke et al., 2020). To describe the shape of the distributions of climate variables, we

turn to the four moments, mean, standard deviation, skewness and kurtosis (Fig. 4, von Storch and Zwiers, 1999). We compute

them for every grid box and time period. These are then area-averaged globally or zonally when providing the respective means

of the moments.

The generalized moments of random variable $X$ of a point $A$ for a sample of size $N$ are defined using the expected value as

$$\mu'_n = \mathbb{E}[(X - A)^n] = \frac{1}{N} \sum_{i=1}^{N} (X_i - A)^n, \tag{1}$$

where $n$ designates the $n$th moment (Papoulis and Pillai, 2002). There are several points of interest that can be substituted for

$A$. One such choice is the origin, that is $A = 0$, such that the $n$th moment around the origin (Papoulis and Pillai, 2002; von

---

[4]Fig. S3 in the Supplement shows the effect of different choices of kernel lengths and compares this method to linear detrending with breakpoints.



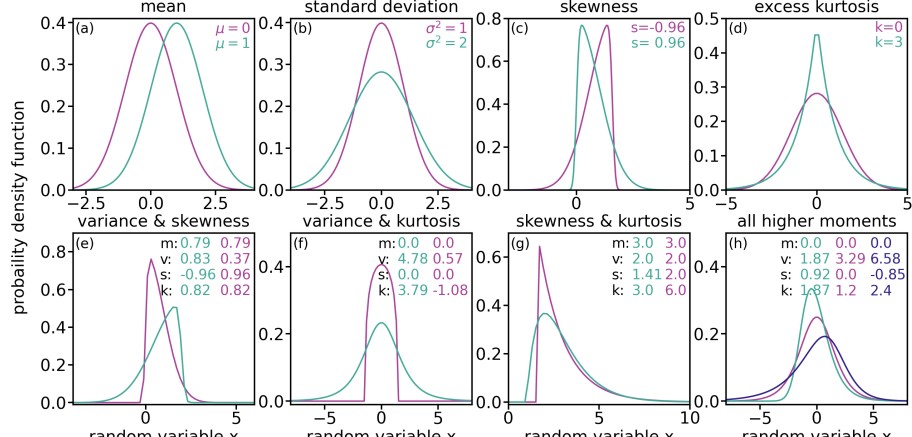

**Figure 4.** Visualization of changes in the moments of the distribution of a random variable with individual changes on top and concurrent changes on bottom. Distributions for a lower (pink) and higher (green) value are shown. Panels (a) – (d) show changes in just one moment: (a) mean, (b) variance, (c) skewness, (d) excess kurtosis. Panels (e) – (h) show exemplary combinations of changes in the higher moments with constant mean: (e) opposite changes in variance and skewness, (f) concurrent change in variance and kurtosis, (g) concurrent change in skewness and kurtosis, (h) changes in all higher moments. Fig. S4 in the Supplement shows exemplary timeseries corresponding to the distributions.

Storch and Zwiers, 1999), sometimes called $n$th raw moment, is

$$\mu_n = \mathbb{E}[X^n] = \frac{1}{N} \sum_{i=1}^{N} X_i^n. \tag{2}$$

Thus, the first raw moment of a distribution is the arithmetic mean $\mu_1 \equiv \mu$ (Fig. 4a, Papoulis and Pillai, 2002).

Considering the moments about the mean $\mu$ instead yields

$$m_n = \frac{1}{N} \sum_{i=1}^{N} (X_i - \mu)^n, \tag{3}$$

called $n$th central moment (Papoulis and Pillai, 2002; von Storch and Zwiers, 1999). For $n = 2$ this corresponds to the variance

$\sigma'^2$. However, in this paper, we use the symmetric unbiased estimator (Filliben and Heckert, 2024) such that

$$\sigma^2 = \frac{N}{N-1} m_2 = \frac{1}{N-1} \sum_{i=1}^{N} (X_i - \mu)^2. \tag{4}$$

It describes the spread of the distribution — the larger the variance and its square root standard deviation $\sigma$, the larger the spread around mean $\mu$ (Fig. 4b).



Moments of order higher than 2 are often normalized using the standard deviation $\sigma'$ (Kenney and Keeping, 1948). The standardized third central moment, skewness $s$,

$$s = \frac{m_3}{m_2^{3/2}}. \tag{5}$$

describes the (a)symmetry of a distribution (von Storch and Zwiers, 1999). Skewness is zero for a symmetric distribution, e.g. for the normal distribution. For negative skewness, the weight of the distribution is at higher values, with mode and median larger than the mean (Fig. 4c). This implies a stronger tail for lower than higher values, the distribution is "skewed left". For positive skewness, on the other hand, mode and median are smaller than the mean, the distribution is skewed towards higher values. Generally, skewness larger than 1 or smaller than $-1$ can be considered high. However, to test whether any skewness found differs significantly from a normal distribution, we test for its deviation from normality using a t-test. For this test, the null hypothesis is that the found skewness and that of a corresponding normal distribution are the same and thus 0. We define the threshold for the p-value to be 0.05.

Lastly, we compute the standardized fourth central moment, kurtosis. For a normal distribution this yields a kurtosis of 3. To derive an estimator which is 0 for normal distributions, kurtosis is often shifted by $-3$ to derive excess kurtosis $k$ (Filliben and Heckert, 2024). Based on the fourth and second central moment, we calculate excess kurtosis as

$$k = \frac{m_4}{m_2^2} - 3. \tag{6}$$

We use excess kurtosis throughout the manuscript, for brevity we will refer to it as kurtosis from now on. Kurtosis captures the heaviness of the tails of a distribution (Fig. 4d). If excess kurtosis is negative, the tails are thinner than those of a normal distribution. Conversely, positive excess kurtosis corresponds to heavier tails. Generally, positive kurtosis and skewness co-occur for datasets with more extreme values (Doane and Seward, 2011). As for skewness, we check again for non-normality using the hypothesis test derived by Anscombe and Glynn (1983) with a threshold for the p-value of 0.05. For all computations, we ignore rare not a number (nan) values in the temperature or precipitation fields.

Changes in moments often occur concurrently and can then both enhance or counteract each other (Fig. 4). For example, a concurrent increase in variance and decrease in skewness can lead to the appearance of a strong low value tail (Fig. 4e). Variance and kurtosis, on the other hand, generally amplify their effects on the tails if both change in the same direction, e.g. if both increase, the tails of a distribution grow on either end (Fig. 4f).

### 3.2 Spectral analysis

In order to analyze how variability of surface temperature and precipitation depend on timescale, we further compute the power spectral density (PSD), also called the power spectrum. If a process contains (quasi-)oscillatory components, the spectrum shows a peak at their periodicity with a certain width related to the damping rate of that process. The spectrum's background and scaling reflect the persistence (or memory) of the process (Ditlevsen et al., 2020).

The auto-covariance function for a random variable $X_t$ at times $t_1$ and $t_2$ is given by the expectation value of its variance as

$$\gamma(t_1, t_2) = \mathcal{E}[(X(t_1) - \mu(t_1))(X(t_2) - \mu(t_2))], \tag{7}$$





where $\gamma(0) = \mathcal{E}[X^2]$ is the variance.

If the timeseries samples an ergodic, weakly stationary stochastic process, the auto-covariance and mean are independent of time and thus depend only on lag $\tau = t_2 - t_1$. Assuming further that the data $X_T$ is an excerpt of a theoretically infinite timeseries such that it is non-zero only for an interval $t \in [-\frac{T}{2}, \frac{T}{2}]$ (Ditlevsen et al., 2020), auto-covariance can be written as

$$\gamma(\tau) = \mathcal{E}((X(t) - \mu)(X(t + \tau) - \mu)) \tag{8}$$

$$= \lim_{T \to \infty} \frac{1}{T} \int_{-T/2}^{T/2} X(t)X(t + \tau)\, dt. \tag{9}$$

The PSD $S$ for frequency $\omega$ is then defined as the Fourier transform $\mathcal{F}$ of the autocovariance

$$S(\omega) = \mathcal{F}(\gamma(\tau))(\omega) \tag{10}$$

$$= \int_{-T/2}^{T/2} \gamma(\tau) \exp^{-i\omega\tau}\, d\tau \tag{11}$$

$$= \lim_{T \to \infty} \frac{1}{T} |\mathcal{F}(X_T(t))(\omega)|^2. \tag{12}$$

The spectra of climate variables sometimes scale consistently across timescales following a power law with $S(\omega) \propto \omega^{-\beta}$, with $\beta$ the so-called scaling coefficient (Fredriksen and Rypdal, 2017; Lovejoy and Varotsos, 2016; Huybers and Curry, 2006; Wunsch, 2003). The scaling coefficient then reflects the persistence of the stochastic process.

To estimate PSDs, we apply the multi-taper method (Thomson, 1982; Percival and Walden, 1993) to the detrended timeseries. For data of finite length, this method reduces spectral leakage by computing separate spectra for orthogonal windows, so-called tapers, and averages the resulting spectra. Here, we use three tapers and estimate chi-squared distributed confidence intervals. We smooth the resulting spectrum and cut off artifacts at the low and high frequency end, such that for a timeseries with a timestep $t_s$ a frequency range of $[2t_s, 1000]$ remains. For comparing the variance of the different time periods across timescales, we further compute the spectral gain following Ellerhoff and Rehfeld (2021) by dividing the spectrum of the LGM and Deglaciation respectively by that of the Holocene.

## 4 Results

We examine changes in variability against a backdrop of a changing mean state, which we examine first (Sect. 4.1). Then, we evaluate temperature moments with respect to their dependence on mean state (LGM, Deglaciation and Holocene), timescale and model complexity (Sect. 4.2). Next, we focus on the forcing dependency by analyzing the influence of ice sheet recon-struction (Sect. 4.3.1), meltwater protocol (Sect. 4.3.2) and volcanism (Sect. 4.3.3) on surface temperature variability. Sections 4.4 and 4.5 repeat the analysis for precipitation. Then, we turn to the power spectra of temperature and precipitation, again considering state- and forcing-dependency, as well as differences related to model complexity (Sect. 4.6). We further compare the temperature spectra to results from the LGM reanalysis.





## 4.1 Mean state changes from LGM to Holocene across the ensemble

Between the LGM and Holocene, all simulations show a mean warming and wetting, as evident by the increasing trends in GMST and GMP towards the Holocene (Fig. 2). Overall, MPI-ESM r1–r7 exhibit the largest temperature difference between the LGM and Holocene with an average increase of $5.6°C$. Among the simulations, the anomaly is largest and the simulated LGM temperature lowest for the simulations with GLAC1-D as the ice sheet reconstruction. In the whole ensemble, LGM cooling is widespread and especially pronounced in the high latitudes on land, with the exception of a few localized hot spots

in a few of the simulations, e.g. an Alaskan warm patch in TraCE-21ka (Fig. S11g, h). Inter-simulation differences are generally larger in the high latitudes, especially in the Northern Hemisphere (Fig. 3b). For precipitation, the picture is more diverse, but in most places and especially over land, a drier LGM is simulated. Some simulations show a locally wetter LGM in the tropics, a phenomenon mostly confined to the oceans. ESMs and GCMs show similar latitudinal profiles, while the EMICs miss some precipitation in the inner tropics as well as the mid-latitude westerlies (Fig. 3c).

## 4.2 State- and timescale-dependency of surface temperature

Analyzing the higher order moments of surface temperature reveals their dependence on timescale and model complexity (Fig. 5, S5). Among the moments, **standard deviation** provides a measure that increases with the spread of the distribution (c.f. Sect. 3.1). The standard deviation of surface temperature and its regional differences decrease towards longer timescales (Fig. 5a, d, g). Most of this decrease occurs between annual and decadal timescales. The only exception to this pattern is the EBM,

which has low standard deviation in all periods. In effect, the other simulations and the EBM become more similar to each other towards longer timescales, although differences remain, in particular towards higher latitudes where standard deviation is highest. Differences between the three periods are similarly concentrated in higher latitudes, especially in the northern polar regions. On annual scales, the Holocene standard deviation is smaller at high latitudes than during LGM and Deglaciation, which are similar to each other. For decadal and centennial scales, on the other hand, the Deglaciation stands out with higher

standard deviation, while Holocene and LGM exhibit more similar levels. The LGMR shows a similar pattern on centennial scales.

As a measure of asymmetry, **skewness** indicates whether weight and tails of a distribution are lopsided (c.f. Sect. 3.1). Positive skewness signals that the weight of the distribution is at lower values with a high value tail. For negative skewness the opposite holds. Globally and across latitudes, skewness of temperature is usually close to zero, indicating little asymmetry

(Fig. 5b, e, h, Fig. S5). The EBM is the exception as it shows pronounced negative skew and thus a low temperature tail on annual timescales. This skew in the distribution shrinks towards longer timescales such that the EBM response is more similar to the other models on centennial scales. On centennial scales, this lack of skewness agrees with the results for the LGMR. In certain latitudinal bands more significant deviations from zero exist. For example, the ESMs show positive centennial and to a lesser degree decadal skewness in the tropics during the Holocene. Looking at the individual simulations, this positive

skewness appears in MPI-ESM r1–6 as well as TraCE-21ka (Fig. S9e, h), but disappears with the addition of volcanic forcing in r7 (Fig. 5h, S7). We find that its origin is mostly oceanic (Fig. S12k, o). Furthermore, TraCE-21ka shows a strong bipolar





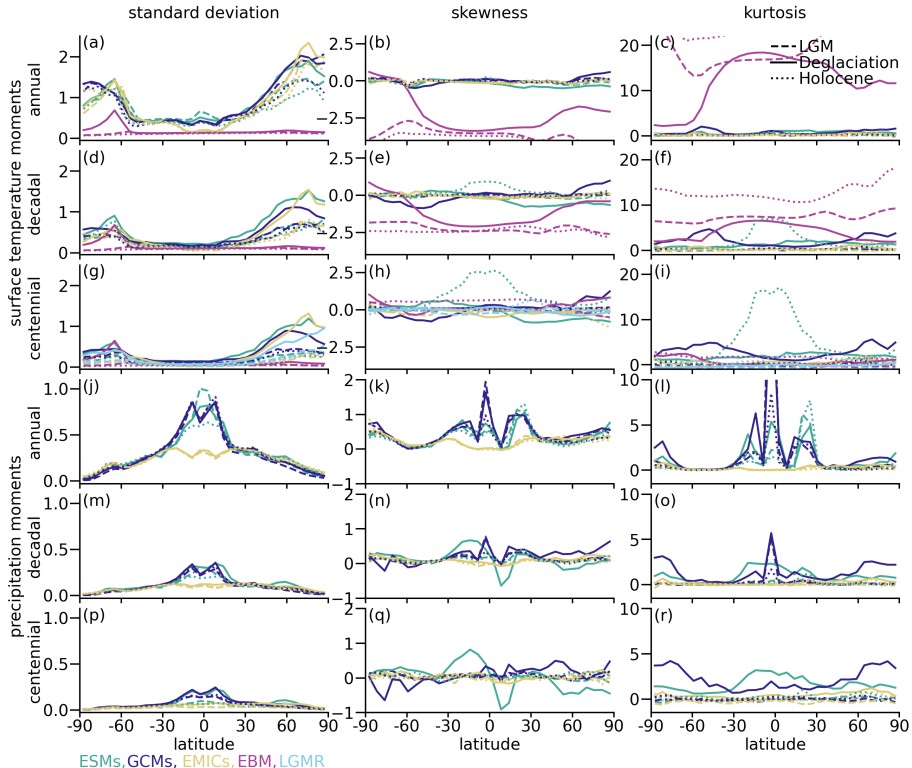

**Figure 5.** Changes of annual, decadal and centennial higher order moments of surface temperature (a-i) and precipitation (j-r) with latitude. For all simulations standard deviation (left column), skewness (middle column) and kurtosis (right column) are shown. Results are differentiated according to period — LGM (dashed), Deglaciation (solid) and Holocene (dotted) — and their complexity — ESMs (green), GCMs (dark blue), EMICs (yellow) and EBM (pink). For centennial temperatures, the LGMR is added in light blue. For temperature the range of skewness and kurtosis of the EBM extend beyond what is shown and for precipitation that of kurtosis in HadCM3B. Fig. S8 in the Supplement shows the full ranges and Fig. S9 the individual simulations.

pattern during the Deglaciation on all timescales with negative skewness in the Southern and positive skewness in the Northern Hemisphere (Fig. 5h and Fig. S12n). All other simulations show either no hemispheric pattern or, in the case of some MPI-ESM simulations and to a lesser degree HadCM3B, the opposite one, although with smaller magnitudes (Fig. 5h and Fig. S12k). For

the MPI-ESM simulations this bipolar pattern mostly shows up in the runs employing the GLAC1-D ice sheet (ch4, r1, r6, r7). The pattern is weaker for the melt-uniform runs (MPI-ESM r4 and HadCM3B r1) and disappears without meltwater forcing (MPI-ESM r3, HadCM3B r2).

    **Kurtosis** reflects the heaviness of the tails, defined here such that positive kurtosis corresponds to tails more pronounced than those of the normal distribution (c.f. Sect. 3.1). Conversely, the kurtosis is negative for less pronounced tails. As for

skewness, the kurtosis is mostly small on annual timescales, across periods and simulations (Fig. 5c, S5c). Towards longer timescales some regional differences emerge, but kurtosis mostly remains small (Fig. 5f, i, Fig. S5f, i). Kurtosis in the LGMR



is small almost everywhere. The ESMs deviate from zero on decadal and centennial timescales in the tropical oceans during the Holocene, where they simulate large positive kurtosis, that is, heavy tails (Fig. 5f, i and Fig. S12x). TraCE-21ka again deviates during the Deglaciation with temperatures that show strong positive kurtosis that is strongest in the high latitudes (Fig. S9o, r and Fig. S12w). The EBM behaves differently already on annual scales, simulating a strong positive kurtosis and thus heavy tails on annual and decadal scales. On centennial scales the EBM is again close to the more complex models with respect to kurtosis. Depending on forcing, the same model setup can simulate a variety of trends and regional patterns (e.g. MPI-ESM r1–4), which merits further investigation.

## 4.3 Influence of forcings on the moments of surface temperature distributions

Using the sensitivity set, we investigate the interaction between forcings and moments of temperature distributions, in particular regarding the underlying ice sheet reconstruction (Sect. 4.3.1), meltwater protocol (Sect. 4.3.2) and volcanic forcing (Sect. 4.3.3).

### 4.3.1 Effect of ice sheet reconstructions on the shape of surface temperature distributions

The spread of the distribution as expressed in standard deviation changes only locally in response to the prescribed ice sheet reconstruction (Fig. 6). On decadal and centennial timescales, ICE6G runs simulate smaller standard deviation in the northern North Atlantic compared to the runs using GLAC1-D (c.f. MPI-ESM r1 and r6, Fig. 6a, b, d, e). This coincides with a reduced sea ice cover in these runs and a smaller degree of warming between the LGM and Holocene (Fig. S1a and c in the Supplement). The opposite pattern occurs in areas of Antarctic sea ice and especially the Weddel sea, where ICE6G runs have higher standard deviation (Fig. 6b, e).

On decadal and centennial timescales, more areas in the simulations using GLAC1-D tend to have significant asymmetry. This manifests in larger skewness, both positive and negative, than in simulations using ICE6G (Fig. 7, Fig. S16). No such pattern exists for annual temperatures (Fig. S17). The trend towards larger areas or magnitudes of negative skewness in simulations using GLAC1-D exists in various areas, for example the Northeastern Pacific, the Arctic ocean and North Atlantic. In the North Atlantic, the difference in negative skewness is particularly pronounced for annual temperature distributions but persists for the Deglaciation also on longer timescales. The ICE6G simulations, including HadCM3B r2, on the other hand, have no such negative skewness in the Arctic Ocean and often simulate positive skew instead, in particular in the Barents and Greenland Sea.

For the Southern Ocean, especially the Weddel Sea, and the Indian ocean off Australia, the GLAC1-D simulations tend towards more positive skewness during the Deglaciation. For the areas of substantial positive skewness along the equator and especially the Indian Ocean and Pacific during the Holocene differences depend on timescale. On centennial scales, the ICE6G simulations show enhanced skewness over larger areas (Fig. 7f, o). On decadal scales, however, the patterns are opposed between the two sets of simulations: r2 with ICE6G has more significant skewness in comparison to r1, whereas r5 with ICE6G has less in comparison to r6 (Fig. S16). MPI-ESM r6 also has such an enhanced skewness pattern on annual scales, while the patterns of the other simulations resemble each other (Fig. S17). This indicates that the differences in parametrization



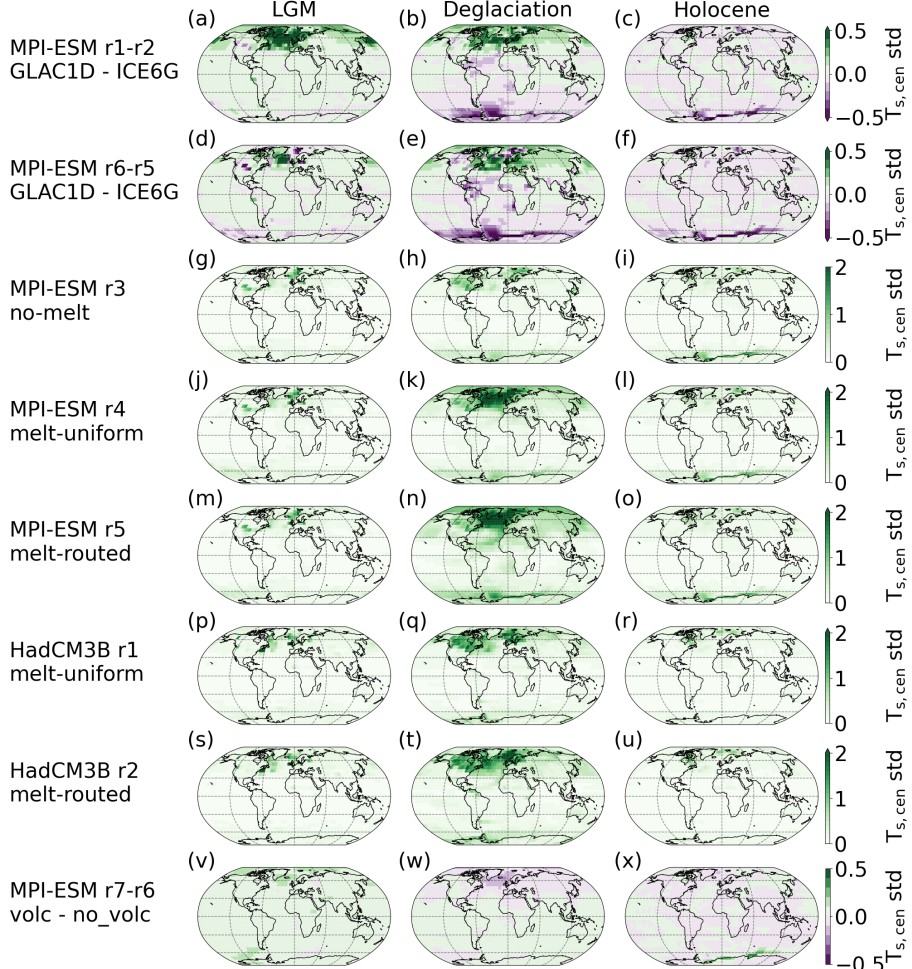

**Figure 6.** Regional effects of forcings on centennial standard deviation of surface temperature. (a) – (f) Influence of ice sheet forcing as differences between MPI-ESM runs using GLAC1-D and ICE6G. (g) – (u) MPI-ESM (g – o) and HadCM3B (p – u) simulations following different meltwater protocols. (v) – (x) Difference between MPI-ESM r7 with volcanic forcing and r6 without it.

influence the skewness. The bipolar pattern of negative skewness in the Northern and positive skew in the Southern Hemisphere that emerges on decadal and centennial timescales is enhanced in the GLAC1-D simulations during the Deglaciation.

The chosen ice sheet reconstruction has a limited impact on temperature kurtosis on all timescales analyzed here (Fig. 8, S18, S19).

### 4.3.2 Effect of meltwater protocols on surface temperature distributions

Meltwater forcing affects the standard deviation of temperature on decadal and centennial timescales in the North Atlantic region. The local meltwater protocol introduces the largest standard deviation that extends furthest south (MPI-ESM r2 vs r3





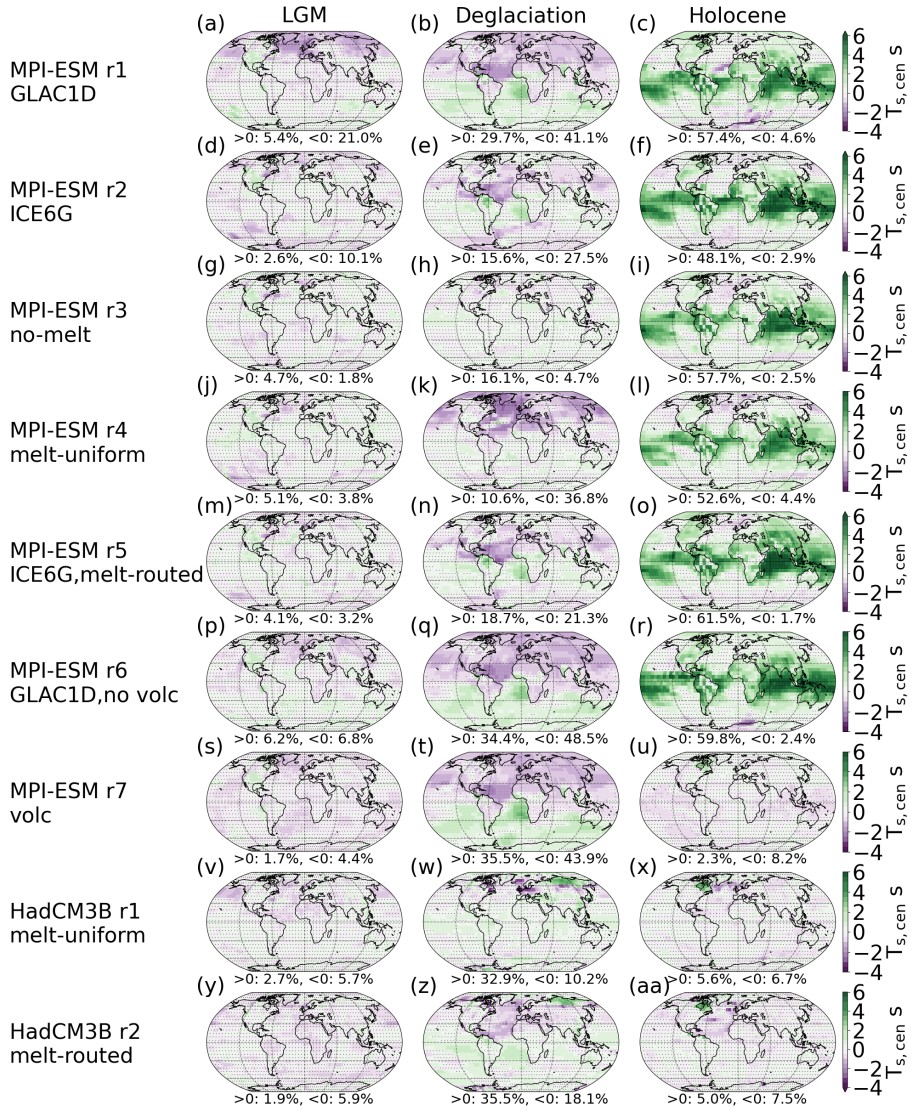

**Figure 7.** Regional effects of forcings on centennial skewness of surface temperature. Forcings are noted along with the run name for each row. Percentages of grid boxes with significant positive and negative deviations from a Gaussian distribution are given. Areas, where changes are non-significant, are hatched.

and r4 and HadCM3B r2 vs. r1, Fig. 6h, k, n, q, t). Local meltwater distribution further has the largest signal in the Southern Ocean. Using the protocol without meltwater removes a source of significant variability, as the comparison of MPI-ESM r3 with r2 and r4 shows (Fig. 6h, k, n).

The deglacial meltwater forcing affects centennial temperature skewness particularly in the North Atlantic (Fig. 7g–o and v– aa): In the melt-routed scenario (MPI-ESM r2, HadCM3B r2), negative temperature skewness is localized there far more than





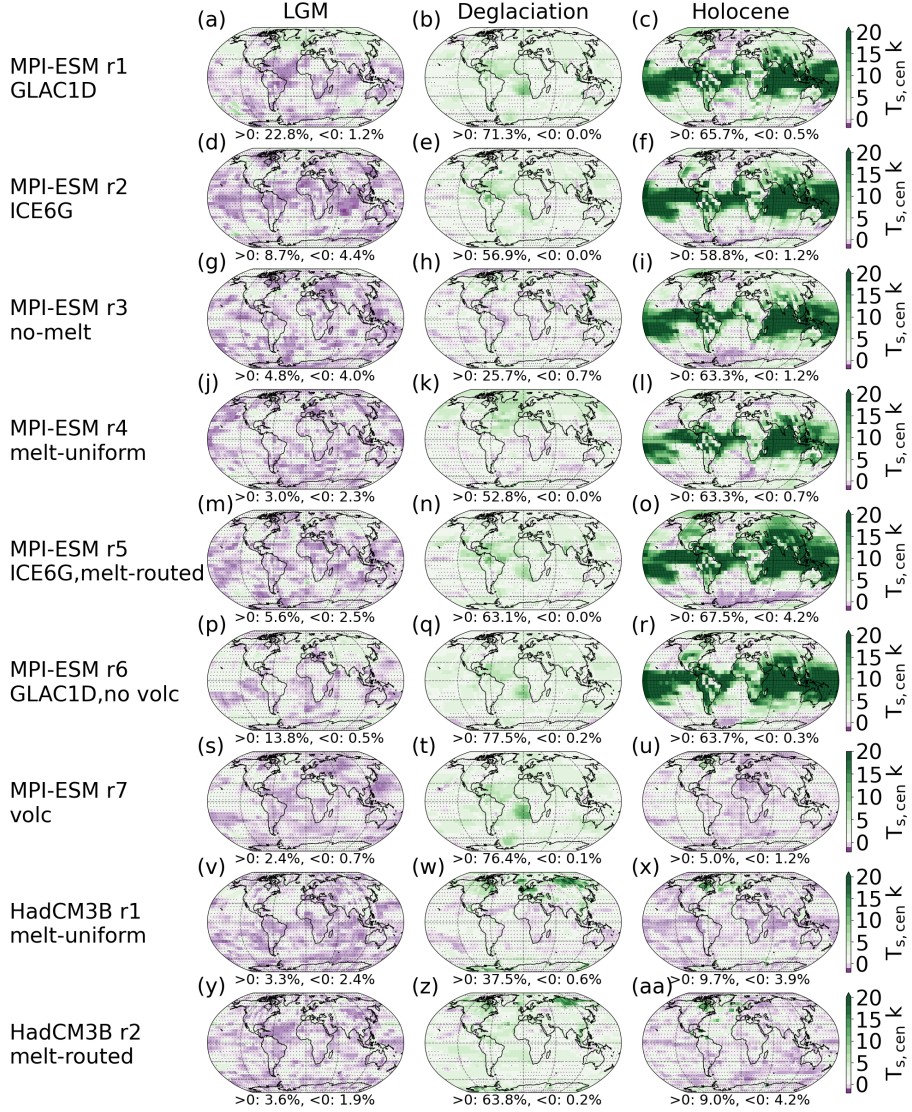

**Figure 8.** Regional effects of forcings on centennial kurtosis of surface temperature. Forcings are noted along with the run name for each row. Percentages of grid boxes with significant positive and negative deviations from a Gaussian distribution are given. Areas, where changes are non-significant, are hatched.

in the uniform scenario (MPI-ESM r4, HadCM3B r1). The melt-routed runs further show positive skewness in the Southern Atlantic and the southeastern Pacific. On the other hand, the uniform runs do not show consistent patterns: MPI-ESM r4 simulates negative skewness over large parts of the middle and high latitudes in the Northern Hemisphere, whereas HadCM3B r1 only has significant skewness in some regions, mostly positive, with the exception of Europe and the Greenland Sea (Fig.





7j–l vs. v–x). The absence of meltwater forcing, as in MPI-ESM r3, results in a notable lack of significant skewness across the globe.

Meltwater forcing has a limited impact on the heaviness of tails in the form of kurtosis (Fig. 8, S12). Across the ensemble, including meltwater mostly introduces a shift to more positive kurtosis during the Deglaciation, especially in the North Atlantic. This positive shift is stronger in the melt-routed (HadCM3B r2, MPI-ESM r2) than the melt-uniform simulations (HadCM3B r1, MPI-ESM r4) on all timescales, although to varying degrees.

### 4.3.3 Effect of volcanism on surface temperature distributions

The effects of volcanism are mostly limited to shorter timescales for standard deviation of temperature (Fig. 6v–x). Even on annual scales it has a limited effect on standard deviation. Globally, the run with volcanic forcing has higher standard deviation during the LGM. For the Deglaciation and Holocene its standard deviation is generally higher in lower latitudes, but smaller at higher latitudes, in particular over the North Atlantic during the Deglaciation. An exception are some areas off the Antarctic coast during the Holocene.

Generally, volcanism results in negatively skewed temperature distributions or a reduction in positive skew since it lowers temperatures after eruptions (Fig. S17j–o). It has the strongest effect on shorter timescales and during the LGM and Holocene. In the LGM, volcanic activity introduces a pronounced negative signal, mostly confined to the tropics. During the Holocene, skewness is decreased as well, turning a strong positive signal over the tropics and most land areas into slightly negative skewness in parts of the tropics and effectively zero elsewhere. On centennial scales, volcanic activity mainly manifests in skewness during the Holocene, where it again counteracts strong positive skewness in the tropics (Fig. 7r, u).

In contrast to ice sheet and meltwater forcings, volcanic forcing impacts kurtosis on all timescales and for all periods (Fig. 8p–u, S18j–o, S19j–o). For annual temperatures, it shifts the kurtosis to be positive in extended areas, in particular in the tropics as well as the northern mid-latitudes. This effect persists on decadal and centennial timescales for the LGM and Deglaciation. During the Holocene and on longer timescales, on the other hand, a low and mid-latitude band of positive kurtosis in the tropics is reduced with the inclusion of volcanic forcing.

### 4.4 State-dependency of precipitation at annual to centennial timescales

The **standard deviation** of precipitation is high in the tropics and decreases towards higher latitudes (Fig. 5). In particular, standard deviation is high over the tropical oceans in the region of the intertropical convergence zone (ITCZ, Fig. 5j and Fig. S14a–i), where mean precipitation is also highest (Fig. 3c). In contrast, standard deviation is not heightened in mid-latitudes, where mean precipitation is enhanced. The tropical band of increased standard deviation exists only to a lesser degree in the EMIC simulations, in which it covers the tropics more homogeneously than in the GCMs and ESMs (Fig. S9 in the Supplement). While there is almost no state-dependency on annual timescales, on centennial scales the spatial patterns of standard deviation differ between the periods. During the Deglaciation, the tropical pattern of enhanced standard deviation remains, albeit less pronounced than on annual scales. The spatial patterns of LGM and Holocene, on the other hand, are more homogeneous and the tropical standard deviation is similar to that at other latitudes (Fig. 9). The only exception is the FAMOUS





simulation, which has enhanced tropical standard deviation for all three periods, as well as the overall largest magnitudes in
the ensemble on centennial scales (Fig. S9p).

The higher moments show more diverse patterns for precipitation (Fig. 5, Fig. S5). For **skewness**, the simulations mostly
show positive precipitation skewness on annual scales, with some negative skewness areas near the equator (Fig. S14r–o). The
tropics also show the largest positive skewness. There are very few differences between LGM, Deglaciation and Holocene
in all simulations (Fig. 5k, S14j–r). The EMICs simulate the smallest skewness, although the positive deviation from zero is
still significant almost everywhere (Fig. S14p–r). The ESMs and GCMs, on the other hand, simulate very similar patterns on
annual scales. Starting on decadal and even more strongly on centennial scales, the patterns diverge for the different periods
and simulations (Fig. 5q, Fig. S13j–r).

During the LGM and Holocene, skewness is close to zero and thus indicates predominantly symmetric distributions. Where
the test indicates significant deviations, these tend towards positive skewness. During the Deglaciation, skewness patterns are
far more diverse, with a larger spread and including negative excursions. The areas of negative skewness are mostly around the
equator, but also sometimes in the high northern (for MPI-ESM simulations with a GLAC1-D ice sheet, Fig. 10) or the high
southern latitudes (for TraCE-21ka, Fig. S13k, n). In a bipolar pattern, TraCE-21ka further simulates high positive skewness
in the high northern latitudes (Fig. S9n, q and Fig. S13n). The EMICs and FAMOUS show almost no significant skew in all
periods on decadal and centennial scales (Fig. S9).

Precipitation **kurtosis** is mostly positive on annual scales in ESM and GCM simulations, in particular in the tropical regions
(Fig. 5, Fig. S5, Fig. S14s–x). LGM and Holocene exhibit no significant kurtosis on longer timescales. During the Deglaciation,
though, positive kurtosis persists, some models even simulate significant positive kurtosis in polar regions. This is, for example,
the case over the North Atlantic in TraCE-21ka (Fig. S13n). The EMICs, on the other hand, have some significant kurtosis only
during the Deglaciation on annual scales and otherwise show no significant deviation from zero in contrast to the more complex
models.

## 4.5  Changes in precipitation distribution shape in response to forcings

The distribution of precipitation and its extremes show little dependence on ice sheet reconstruction, meltwater protocol or
volcanism (Fig. S20 – S25). A notable exception are centennial moments during the Deglaciation, all of which reveal a depen-
dence on meltwater forcing (Fig. 9, 10 and 11). For **standard deviation**, the melt-routed runs (MPI-ESM r2 and HadCM3B
r2) exhibit higher values in the tropics and North Atlantic than the melt-uniform ones (MPI-ESM r4 and HadCM3B r1). The
lack of meltwater in the no-melt simulation (MPI-ESM r3) reduces standard deviation in these areas as well.

Only in some areas of large mean change is there a change in skewness (Fig. 10) and kurtosis (Fig. 11). For **skewness** at
centennial scales, the ice sheet reconstruction predominantly affects precipitation distributions in the tropics, North Atlantic
zone, as well as mid- and high latitudes in Eurasia (Fig. 7). These are also the areas with most changes in temperature skewness
in general. The precipitation skewness in the tropics generally follows the same pattern for both ice sheet reconstructions, but
with both positive and negative skewness enhanced in the GLAC1-D simulations. An exception are high northern latitudes





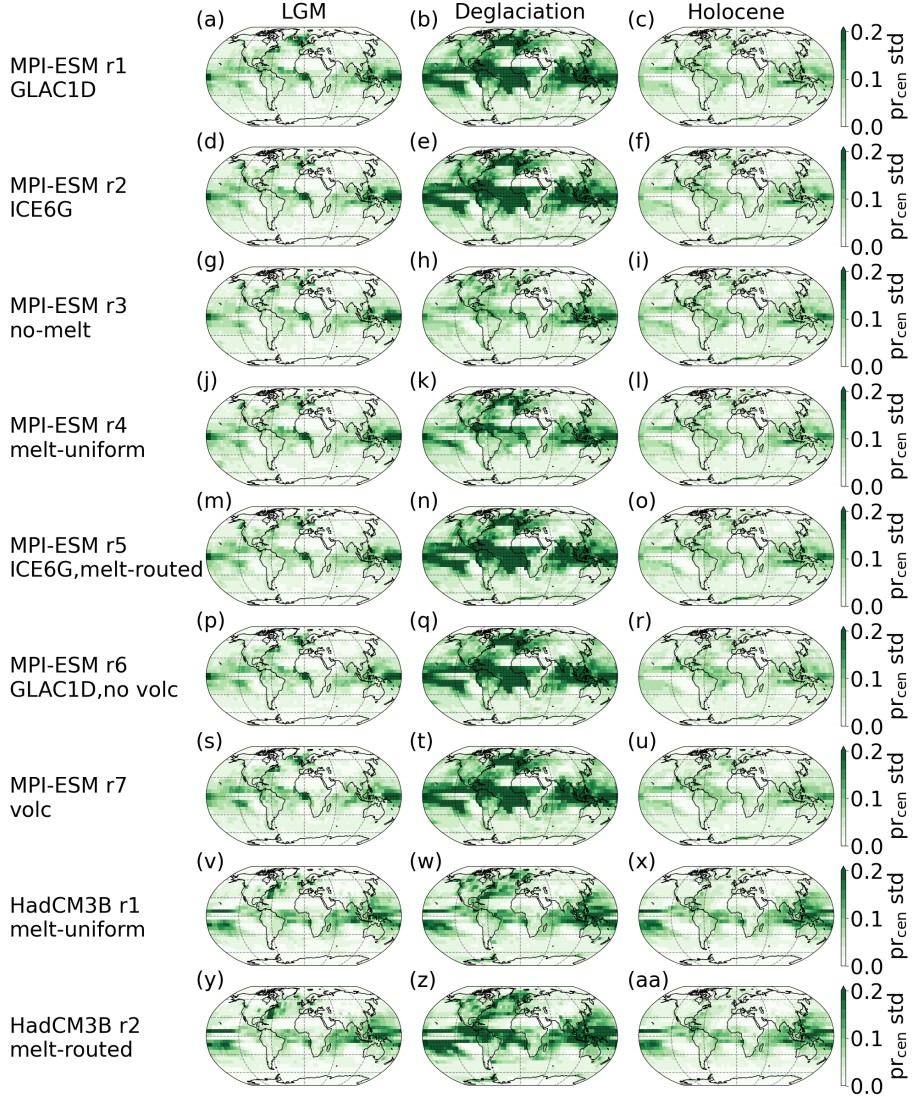

**Figure 9.** Regional effects of forcings on centennial standard deviation of precipitation. Forcings are noted along with the run name for each row.

during the Deglaciation, where skewness is positive in the ICE6G simulations and negative in the GLAC1-D ones (Fig. 10, Fig. S22).

Injecting meltwater introduces significant skewness in precipitation distributions during the Deglaciation on centennial
timescales which does not exist without it (c.f. no-melt scenario in Fig. 10h). Both melt-routed simulations (MPI-ESM r5 and HadCM3B r2) have a signal of negative skewness in the eastern equatorial pacific, although it extends further west in MPI-ESM with a positive signal to the south of it. Only the positive signal somewhat remains in the melt-uniform runs. These





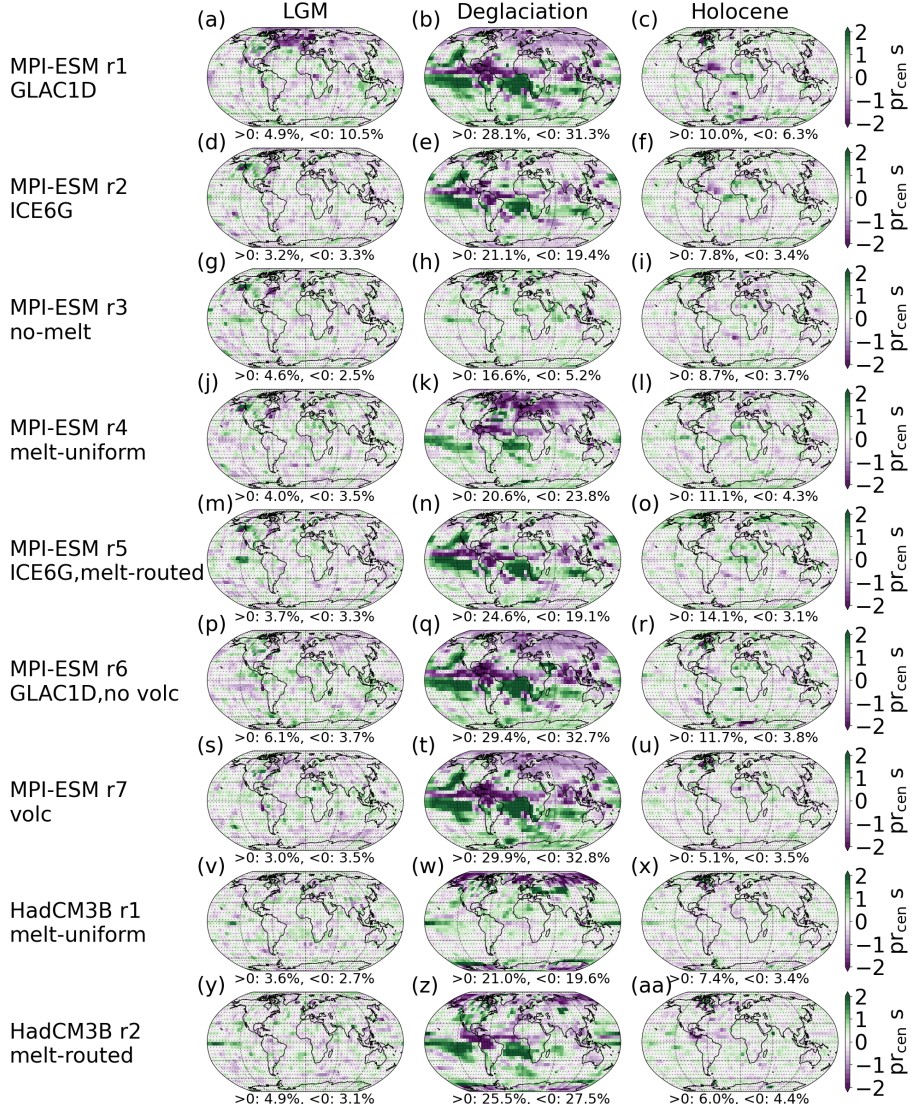

**Figure 10.** Regional effects of forcings on centennial skewness of precipitation. Forcings are noted along with the run name for each row. Percentages of grid boxes with significant positive and negative deviations from a Gaussian distribution are given. Areas, where changes are non-significant, are hatched.

runs further exhibit strong negative skewness in high northern latitudes, although in different areas. The simulations also differ in skewness in the Southern Hemisphere, but without any unifying patterns between HadCM3B and MPI-ESM runs.

The influence of meltwater forcing on centennial **kurtosis** during the Deglaciation shows up predominantly as positive kurtosis, with no-melt (MPI-ESM r3) having the fewest areas with significant positive kurtosis. Melt-routed runs have more positive kurtosis in the tropics, whereas melt-uniform simulations have more in the high northern latitudes. In some other areas,




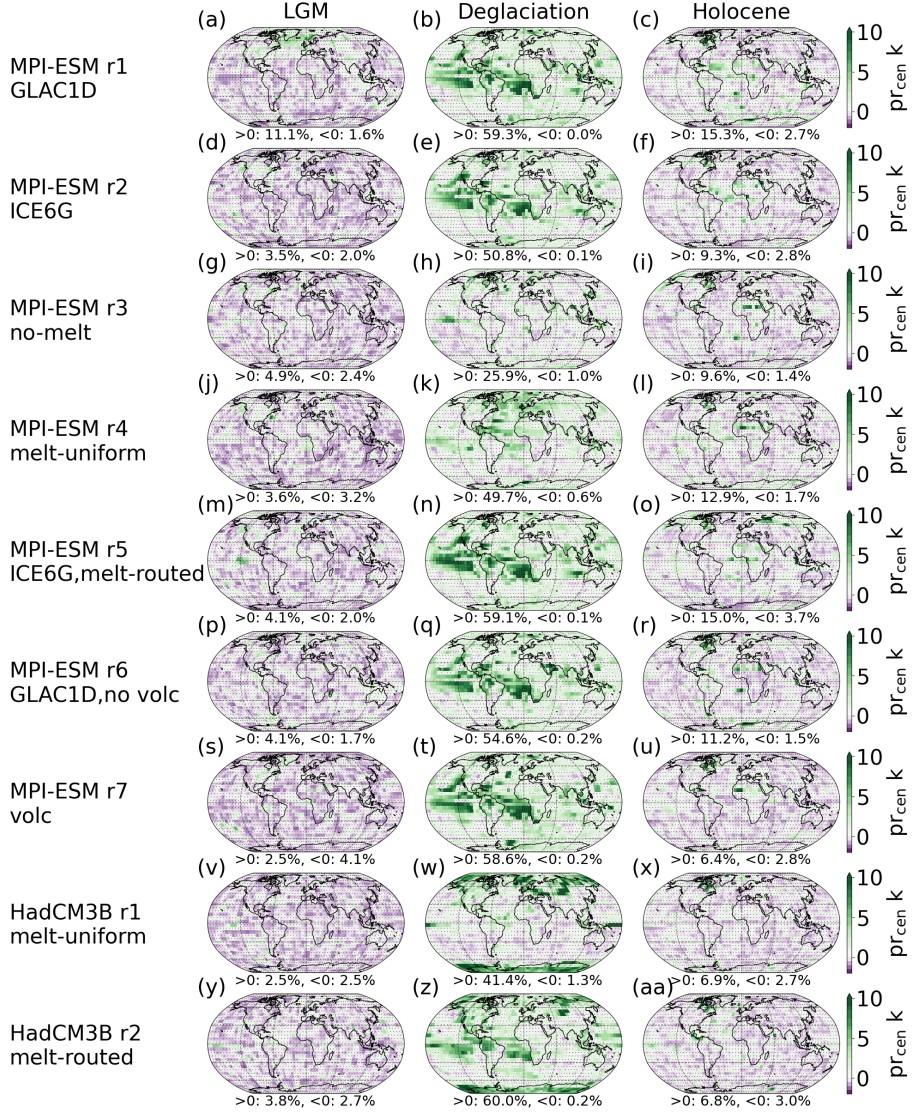

**Figure 11.** Regional effects of forcings on centennial kurtosis of precipitation. Forcings are noted along with the run name for each row. Percentages of grid boxes with significant positive and negative deviations from a Gaussian distribution are given. Areas, where changes are non-significant, are hatched.

trends in MPI-ESM and HadCM3B are opposite with respect to meltwater forcing. In HadCM3B there are also differences on decadal scales, but those are entirely confined to the eastern equatorial Pacific, where r2 (melt-routed) has more extensive
positive kurtosis than r1 (Fig. S24q, t). Ice sheet reconstruction, meltwater protocol and volcanic forcing usually affect the moments of temperature more than those of precipitation.



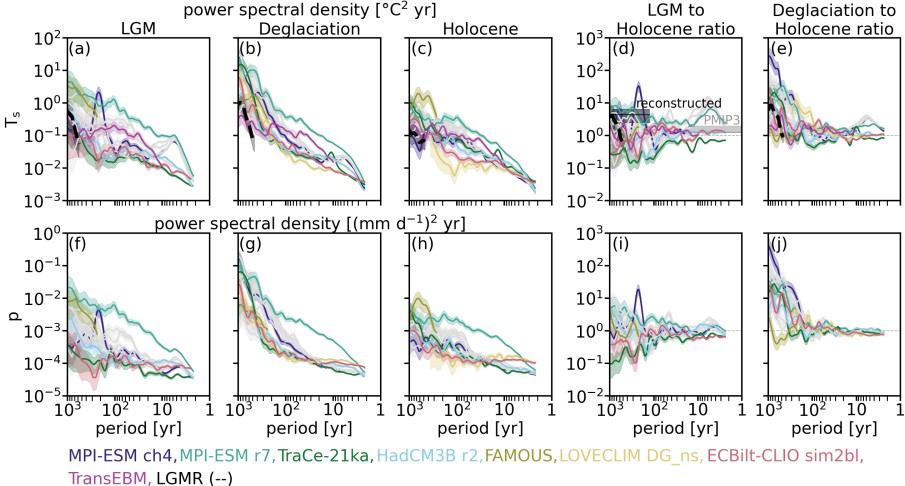

**Figure 12.** Spectra and spectral ratios of surface temperature (top row) and precipitation (bottom row) variability with chi-squared distributed confidence intervals. Both x- and y-axis are shown on a logarithmic scale. The spectra are separated by time period with (a, f) LGM, (b, g) Deglaciation and (c, h) Holocene. The spectral ratios highlight the differences between the periods showing the LGM-to-Holocene (d, i) as well the Deglaciation-to-Holocene ratio (e, j). The sensitivity set (here in gray) is shown in Fig. S27 in the Supplement. In panel d, Rehfeld et al. (2018)'s estimated range of the multi-centennial to millennial LGM-to-Holocene variance ratio based on proxy reconstructions is marked for comparison.

## 4.6 Spectral analysis of the variability of surface climate

To add to the analysis of variability, we examine the spectra of surface temperature during the LGM, Deglaciation and Holocene with respect to dependence on state- and timescale (Sect. 4.6.1), forcing (Sect. 4.6.2) and complexity (Sect. 4.6.3). We fur-
ther compare the simulated spectra to those of the LGM Reanalysis (Sect. 4.6.4). Lastly, we present the spectral analysis of precipitation (Sect. 4.6.5).

### 4.6.1 Surface temperature spectra since the LGM

Generally, we find temperature spectra that increase towards longer timescales, some of which level off at multi-centennial scales (Fig. 12). This pattern is particularly strong for the Deglaciation, where it can also be found across latitudinal bands
(Fig. S28 middle column). However, the regional spectra can be flat during the LGM and Holocene, for example in the tropics or mid-latitudes after a scale break at multi-decadal scales. The spread between the simulations increases towards longer timescales, especially during the Deglaciation and Holocene. During the LGM, all MPI-ESM simulations show increased variability with a broad peak on inter-annual scales, such that the spread is comparatively large there. This increased variability originates in the tropical regions (Fig. S28j, m) and relates to the simulated El-Niño Southern Oscillation. In the tropics, the
spread between simulations remains similar across timescales in all three periods.





We use the PSD ratios to investigate the state-dependency of time-varying variance (Fig. 12d, e). The ratios between LGM and Holocene depend on simulation and timescale. In some simulations, the LGM has larger PSD across all timescales (e.g. MPI-ESM r7), for others it is the Holocene (e.g. TraCE-21ka). For most simulations, it changes with timescale as many have larger Holocene spectral power on centennial scales, but less on decadal and millennial scales (Fig. 12d).

With very few exceptions, the deglacial spectrum contains the largest power, especially above centennial timescales (Fig. 12e). This pattern mostly holds for regional spectra across latitudinal bands (Fig. S28, S29). This is partially because the increase in power from inter-annual to millennial timescales is steepest during the Deglaciation, whereas the scaling is smaller for LGM and Holocene. MPI-ESM ch4 shows the largest ratio between Deglaciation and Holocene, whereas FAMOUS has the smallest due to it almost always having more power during the Holocene than the other simulations (Fig. 12c, e).

### 4.6.2 Forcing-dependency of the temperature spectrum to ice sheet reconstruction, meltwater forcing and volcanism

The temperature spectra vary significantly in magnitude and pattern between simulations and in response to external forcing differences (Fig. 12, Fig. S28). For GLAC1-D simulations, we find increased variability during the LGM on decadal to centennial timescales (c.f. r1 & r6 vs. r2 & r5), mainly in the Northern Hemisphere mid- and polar latitudes.

Meltwater forcing, on the other hand, has the strongest impact during the Deglaciation, again for the northern high latitudes.
For the global spectra there is little difference between runs using the melt-routed (MPI-ESM r2 and HadCM3 r2) versus melt-uniform protocol (MPI-ESM r4 and HadCM3 r1). However, the impact of meltwater forcing can be seen in comparison to the run without meltwater forcing, MPI-ESM r3. Among MPI-ESM r1–r7, this run has the lowest variability during the Deglaciation.

Volcanism has a particularly strong impact on the temperature spectrum. MPI-ESM r7 has the largest PSD from inter-annual
up to centennial timescales during all three periods, even in comparison to r6, from which it differs only in the inclusion of volcanic forcing. While the other MPI-ESM runs show a drop in PSD on inter-annual scales (especially during the LGM, see Fig. 12a vs. Fig. S27a), r7 shows consistent increase in variability until at least centennial scales. On longer timescales, too, it is on the upper end of simulated variability.

### 4.6.3 Dependence of the spectral power of temperature on model complexity

For the most part, GCMs and EMICs display less spectral power than the ESMs up to multi-centennial scales, where most show an increase in variability. Sometimes, MPI-ESM ch4 is the exception to this rule. For example it agrees with the other MPI-ESM simulations on inter-annual scales during the LGM, but then is more similar to the non-MPI-ESM simulations on multi-decadal scales. It further exhibits a strong $200 - 400$ year periodicity during the LGM that is absent in all other simulations. This signal originates in the Southern Hemisphere sea ice, grows stronger towards higher latitudes and extends
into the tropic (Fig. S28, Fig. S36d, S37 Sec. S7.1 in the Supplement). The spectral power of the EBM is at the higher end on decadal to centennial timescales. There, MPI-ESM r7 is often the only simulation with more power. However, it levels off around centennial scales, with only moderate increases in variability afterwards, such that its variability is among the lowest on millennial scales, indicating a lack of persistence.





### 4.6.4 Comparison of the simulated surface temperature variability to the LGM reanalysis

The magnitude of the LGMR power spectrum generally falls within the range of the ensemble. During the LGM it shows similar levels of variability as the GCMs, matching their increase towards millennial scales. For the Deglaciation, it starts at the lower end of variability, but again exhibits a strong increase. This increase suggests a larger scaling in comparison to the ensemble, however, confidence in statements about scaling in the LGMR would require investigating a larger span of timescales in it. For the Holocene, the LGMR is at the lower end of spectral power. However, it is also mostly below the range of LGM

to Holocene spectral ratios found in reconstructions by Rehfeld et al. (2018) (Fig. 12d). Most simulations also fall below that range, especially on decadal to centennial timescales. Above centennial scales, many of the runs among MPI-ESM r1–r7 are in agreement with it. Notably, MPI-ESM r7 agrees with the range found by Rehfeld et al. (2018) on most timescales, even if is at the lower end for multi-decadal scales.

### 4.6.5 Dependency of precipitation spectra on state, timescale, model complexity and forcing

With the exception of MPI-ESM r7, global spectra are quite flat across short timescales, then feature an increase starting on multi-decadal (MPI-ESM r1–r6) or centennial scales (the remainder of the ensemble). This increase levels off again around millennial scales. For the Deglaciation, the increase is very sharp, whereas it is smaller and often more gradual during the Holocene and LGM. Thus, the variability is found during the Deglaciation above centennial timescales, with FAMOUS as the only exception. Regionally, LGM and Holocene spectra can be white especially in the tropics, with only some simulations

showing an increase in variability towards longer timescales for mid- and polar latitudes (Fig. S29).

The MPI-ESM simulations generally have larger precipitation variability during the LGM than the Holocene, whereas all other models show the opposite (Fig. 12i). All simulations show larger precipitation variability during the Deglaciation than during the Holocene with the difference increasing towards longer timescales (Fig. 12j). Precipitation variability is among the highest in the MPI-ESM and lowest in the EMIC simulations, globally and regionally (Fig. S30).

Inspecting the effect of forcings in the spectra of the sensitivity set reveals similar relationships as for the temperature spectra. Using GLAC1-D leads to larger LGM variability on on multi-decadal and longer timescales for mid- and high northern latitudes (Fig. S31). The no-melt protocol shows a distinct lack of deglacial variability, especially in the Northern Hemisphere. In all periods, MPI-ESM r7 with its volcanic forcing has significantly larger variability than all other simulations on interannual to centennial timescales. This difference is even more pronounced than for the temperature spectra and reaches up to one order of

magnitude. Regionally, too, the spectral power of MPI-ESM r7 is always on the upper end such that it stands out even among the MPI-ESM simulations.





## 5    Discussion

We investigate variability changes before, during and after a period of global warming in an ensemble of transient simulations of the Last Deglaciation across a climate model hierarchy. Among them, variability differs considerably (c.f. Table 2). The differences we find depend on

– **timescale**: Surface temperature shows a decrease in standard deviation, larger absolute skew and an increase in kurtosis towards longer timescales (Sect. 4.2). For precipitation, standard deviation decreases with timescale (Sect. 4.4). During the LGM and Holocene, skewness and kurtosis of temperature and precipitation similarly show a decrease with timescale. During the Deglaciation, however, precipitation skewness and kurtosis show changes in spatial patterns that lead to regionally heterogeneous trends.

– **background state**: Generally, the state-dependency of surface temperature increases with timescale for all moments (Sect. 4.2). For standard deviation, the Deglaciation has the largest values overall among the periods we examine. During the LGM, skewness and kurtosis deviate little from zero in contrast to both Deglaciation and Holocene, which exhibit prominent spatial patterns. For precipitation, trends differ between moments and are more complex (Sect. 4.4). EMICs show almost no state-dependency unlike ESMs and GCMs. For those more complex models, some state-dependency exists on all timescales and for all moments.

– **forcings**: Simulations that differ only by ice sheet reconstruction diverge most on long timescales, although differences can be found even for annual variability (Sect. 4.3.1). Clear patterns emerge for the Deglaciation as the simulations using GLAC1-D have larger standard deviation in the Northern Hemisphere, are generally more skewed and have more areas of significant positive kurtosis. LGM and Holocene show only small differences with ice sheet reconstruction. For precipitation, a difference in the employed ice sheet reconstructions mainly affects skewness, with more skewed distributions for GLAC1-D on longer timescales (Sect. 4.5).

As for ice sheet forcing, the chosen protocol for meltwater injections primarily affects the moments on multi-decadal and longer timescales (Sect. 4.3.2). On these, any kind of meltwater will increase the standard deviation for both temperature and precipitation, with the largest values for routed meltwater. Meltwater forcing also introduces more skewness and larger positive kurtosis. For temperature, these trends manifest as localized negative temperature skewness and enhanced kurtosis in the North Atlantic as the main area of meltwater injection. For precipitation, routed meltwater results in increased positive and negative skew as well as positive kurtosis around the equatorial Atlantic and Eastern Pacific Ocean (Sect. 4.5).

The presence of volcanic forcing primarily affects the moments of temperature with little effect on those of precipitation (Sect. 4.3.3, Sect. 4.5). Its presence creates a low temperature tail as indicated by increasingly negative skewness and decreased positive skewness as well as mostly larger kurtosis. For both temperature and precipitation, volcanic forcing increases spectral power on all timescales and inter-annual to centennial ones in particular.





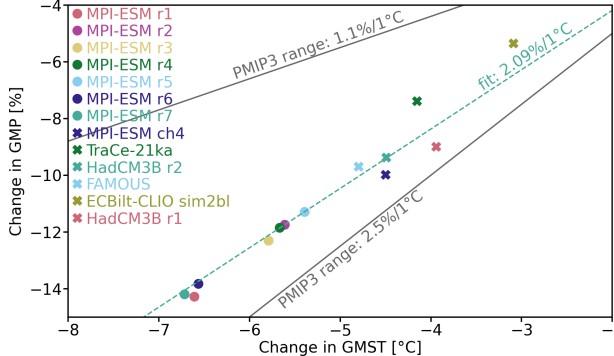

**Figure 13.** Hydrological sensitivity during the LGM and Holocene: percentage change in LGM GMP from Holocene against the change in GMST. The line indicates a 2% change in precipitation per degree of temperature change. The data is fitted linearly with intercept 0.

- **model complexity**: There are substantial differences in simulated variability between categories of model complexity as well as models of similar complexity (Sect. 4.2, 4.4). Except for the standard deviation of temperature, EMICs simulate very little change between states and mostly have higher moments close to zero. In this respect, they differ strongly from ESMs and GCMs, which show more complex spatial patterns.

In the following, we discuss these findings in more detail in the context of existing literature.

## 5.1 Large range in simulated and reconstructed mean changes

The simulated LGM to Holocene changes in GMST range from 3.0 to 6.6°C (Table 2 and S1 in the Supplement). Proxy-based reconstructions and data assimilation approaches provide similar large ranges. Among more recent estimates, Osman et al. (2021) suggest a warming of $7.0 \pm 1.0$°C from the Deglaciation onset to PI, while Tierney et al. (2020) estimate a temperature difference of 6.1°C (5.7, 6.5). On the other hand, Annan et al. (2022) propose $4.5 \pm 0.9$°C and Shakun and Carlson (2010) reconstruct a minimal warming of 4.9°C for the LGM at 22 kyr BP relative to the Altithermal at around 8 kyr BP. While some of the differences can be explained by different reference periods, uncertainty around the level of warming remains and is reflected in the ensemble. Agreement is larger with respect to spatial patterns of warming, with larger changes in the Northern than in the Southern Hemisphere, towards higher latitudes in both hemisphere and over land and areas of melting ice sheets. The temporal pattern of GMST change, on the other hand, differs a lot between simulations. This includes but is not limited to the onset and termination of deglacial warming as well as the timings of periods of abrupt change. These differences in mean state also translate into differences in variability around that mean between simulations, further complicating the search for common mechanisms of variability change.

Precipitation variability since the LGM is less studied than temperature and even fewer proxy reconstructions of hydroclimate exist. There is also no data product like the LGMR, which can be used for comparison of spatial patterns. Therefore, we consider simulated hydrological sensitivity to contextualize our results (Fig. 13). The Clausius-Clapeyron relation estimates a





7% change in saturation water vapor per degree temperature change. However, this is further constrained based on the surface-energy balance and its effect on evaporation, as well as water availability, such that a realistic range for precipitation change per degree temperature change is $1 - 4\%$ (Li et al., 2013). Here, we find a hydrological sensitivity of about 2.09% per degree change in GMST. This agrees with the ranges given for CMIP5/PMIP3 equilibrium simulations by Li et al. (2013) of $1.5 - 3\%$ per degree K and Rehfeld et al. (2020) of $1.1 - 2.5\%$ per degree K.

## 5.2 Increasing state-dependency of variability with timescale

### 5.2.1 Increased standard deviation and spectral power of surface temperature during the deglacial transition

For all moments of surface temperature, we find that state-dependency generally increases with timescale. Simultaneously, the ensemble simulates a reduction in the spread of temperature distributions towards longer timescales with the overall largest values during the Deglaciation. On annual scales, areas of large standard deviation often exhibit large mean changes, too. For

longer timescales, however, spatial patterns of standard deviation change differ from those found in the mean. At decadal and centennial scales, standard deviation is largest over the high latitude oceans, particularly in areas with seasonal sea ice cover and for the Deglaciation in the North Atlantic in response to the changes in the Laurentide ice sheet. As the sea ice cover shrinks with warming, and increases during periods of abrupt cooling, its standard deviation is larger during the Deglaciation than the more stable LGM and Holocene (Fig. S33 in the Supplement). We further find that sea ice influences variability beyond the

interannual to centennial scales as standard deviation of surface temperature shows the imprint of changes in centennial sea ice cover. This importance of sea ice for local variability on decadal and longer timescales is in line with results from Ellerhoff et al. (2022).

The ratio of LGM to Holocene variance mostly shows higher LGM variance with values between 1 and 2 (Tables 2 and S1 in the Supplement). TraCE-21ka is the only simulation with higher Holocene than LGM variance on all investigated timescales.

While some simulations have higher Holocene variance on one of the investigated timescales, none show it across all. The ratios of surface temperature confirm these results. Higher LGM than Holocene variance holds especially true for longer timescales, but with differences depending on timescale (Fig. 12a, c, d). These results resemble those of Rehfeld et al. (2018) who found ratios between 1 and 3 on inter-annual to decadal scales based on CMIP5/PMIP3 equilibrium simulations. Similarly, Shi et al. (2022) conclude that inter-annual temperature variance in PMIP3/4 LGM simulations is 20% higher than in PI simulations

as a consequence of an increased meridional temperature gradient, in particular at mid-latitudes. While we find an enhanced meridional temperature gradient during the LGM at mid-latitudes, our results do not confirm such a large-scale increase of the gradient nor a correlation to the variance ratio (Fig. S38).

Few model-data comparisons of variance exist that include data covering the LGM. When comparing the LGM and Holocene, proxy reconstructions show a LGM variance that is globally about 4 times higher on timescales from 500 to 1750 years (Re-

hfeld et al., 2018). The changes in variance between the LGM and Holocene we find here are thus smaller than proxy records suggest (Table 2), in line with Rehfeld et al. (2018), at least for ratios up to the centennial scale. Simulated LGM and Holocene spectra resemble each other in their general shape, but as outlined above, we find state-dependent features (Fig. 12). This is





in contrast to the lack of state-dependency between global spectra of equilibrium LGM and PI simulations found by Ellerhoff et al. (2022). Since the differences are especially apparent on longer timescales, this might point towards long-term memory

effects missing in equilibrium simulations.

The Deglaciation shows enhanced levels of variance in comparison to the LGM and Holocene on decadal and centennial timescales (Fig. 5d, g, Fig. S6) as well as larger spectral power above centennial scales (Fig. 12e). Northern high latitudes are the largest source of this state-dependency, with further significant state-dependency in high southern and northern mid-latitudes (Fig. 5d, g). This reflects the dynamic nature of the Deglaciation with ice sheets melting, resulting freshwater input

and subsequent reorganization of the climate system. Especially on millennial scales the enhanced variability found in the spectrum matches the oscillatory behavior that Clark et al. (2012) described on this timescale.

Comparing simulations and reconstructions, Zhu et al. (2019) argue in favor of agreement between simulated and reconstructed temperature variability on the global scale based on an analysis of the temperature spectrum of the full timeseries. That analysis considers long proxy records and reconstructions reaching back up to $5\,\mathrm{Myr}$ BP as well as three simulations

used in this paper, TraCE-21ka, LOVECLIM DG_ns and ECBilt-CLIO. With respect to agreement on the global scale, our results agree when comparing simulations to the LGMR. On local and regional scales, however, climate models have repeatedly been found to underestimate variance on longer timescales (Laepple et al., 2023). Similarly, we here find notable differences between regional patterns in the LGMR and the simulations (c.f. Fig. S26). Further, as a reanalysis product, the LGMR uses model simulations as priors and thus might be affected by a lack of variability in models.

Considering the longer timescales included in the global spectra, some of the MPI-ESM runs do fall into the range suggested by the reconstructions (Fig. 12). The LGMR falls partially into the range of other reconstructions and only close to millennial timescales. It shows a steep increase in LGM spectral power towards longer timescales, which translates into an increasing LGM-to-Holocene ratio that is unlike that of most simulations in the ensemble. However, proxy records covering both LGM and Holocene are sparse and both Shi et al. (2022) and Rehfeld et al. (2018) suggest considerable spatial heterogeneity in the

variance ratios. The sparse sampling of variance around the globe is likely to bias the LGMR and thus the comparison. While the LGMR thus exhibits more similar levels of variability to most of the ensemble, comparison to the other reconstructions suggests a lack of simulated regional variability on multi-decadal timescales and beyond.

The difference between the LGMR and other reconstructions leaves uncertainty around the models' abilities in capturing climate variability and thus a potential lack of variability in future simulations with consequences for projected changes in

the frequency and intensity of extremes. To decrease this uncertainty, our findings can provide a basis for more in-depth model-data comparisons of simulated variability during periods of warming. Realizing the full potential of such an analysis requires an ensemble of coordinated experiments using common protocols (as for some of the simulations here) and improved reconstructions of past variability for comparison. This would enable confident identification of which simulations better reflect actual variability of past climate.





**Table 2.** Summary of LGM-to-Holocene changes in the moments. For every moment and timescale, the values according to model complexity are listed as ESMs | GCMs | EMIC | EBM. For the mean, the absolute value of the difference is listed, for the other moments the ratios. Ratios are first computed for the individual simulations (c.f. Tables S1 and S2 in the Supplement for surface temperature and precipitation respectively) and then averaged by category. For EMICs this only includes ECBilt-CLIO here. Note that on centennial scales the ESM and GCM categories include more simulations (MPI-ESM r3, r4 and FAMOUS) than on the annual and decadal scale, leading among other things to a difference in average mean change. It shows that centennial Holocene skewness of ECBilt-CLIO is very close to zero, which produces a very large EMIC ratio. Very large ratios, as for the skewness of centennial precipitation distributions, are a result of moments very close to zero for the LGM and Holocene.

| | $\Delta m_{hol-lgm}$ | | | | $v_{lgm}/v_{hol}$ | | | | $s_{lgm}/s_{hol}$ | | | | $k_{lgm}/k_{hol}$ | | | |
| --- | --- | --- | --- | --- | --- | --- | --- | --- | --- | --- | --- | --- | --- | --- | --- | --- |
| | ESM | GCM | EMIC | EBM | ESM | GCM | EMIC | EBM | ESM | GCM | EMIC | EBM | ESM | GCM | EMIC | EBM |
| $T_{ann}$ | 5.81 | 4.44 | 3.95 | 4.12 | 1.75 | 1.27 | 1.16 | 1.27 | 0.07 | -2.57 | 0.87 | 0.90 | 0.51 | 0.88 | 0.38 | 0.60 |
| $T_{dec}$ | 5.81 | 4.44 | 3.95 | 4.12 | 1.43 | 1.25 | 1.03 | 1.39 | 1.76 | 3.44 | 0.76 | 0.98 | 0.26 | 0.29 | 0.09 | 0.60 |
| $T_{cen}$ | 5.76 | 4.10 | 3.95 | 4.12 | 1.52 | 1.50 | 0.74 | 1.56 | 0.25 | 0.80 | 0.84 | -0.01 | -0.26 | -0.25 | -0.17 | -0.29 |
| $p_{ann}$ | 0.34 | 0.24 | 0.25 | | 1.17 | 0.82 | 0.93 | | 1.08 | 1.09 | 1.36 | | 1.10 | 1.63 | 1.36 | |
| $p_{dec}$ | 0.34 | 0.24 | 0.25 | | 1.10 | 0.82 | 0.93 | | 1.02 | 1.14 | 1.33 | | 0.81 | 1.31 | -0.13 | |
| $p_{cen}$ | 0.34 | 0.22 | 0.25 | | 1.07 | 0.84 | 0.88 | | 0.64 | 1.01 | 101.84 | | -0.22 | -8.97 | -2.44 | |
| LGMR | 6.80 | | | | 1.21 | | | | -0.89 | | | | -0.47 | | | |

### 5.2.2 Larger absolute surface temperature skewness and kurtosis towards longer timescales

Temperature skewness and kurtosis, describing the asymmetry and heaviness of the tails of a distribution respectively, deviate more from zero towards longer timescales, indicating more non-Gaussian distributions and changes in extremes (Fig. 5, Fig. S5). During the Deglaciation, mid- and high latitudes show enhanced values of skewness and kurtosis in ESMs and GCMs. In the Holocene, increased skewness and kurtosis can be found in the tropics for MPI-ESM r1–6 and TraCE-21ka. This pattern of increased tropical skew and kurtosis for the Holocene is notably absent in MPI-ESM ch4 (using a different parametrization than r1–6) and r7 (which includes volcanic forcing) as well as the HadCM3B runs. During the LGM, on the other hand, there is little skew and kurtosis.

The LGMR contains some enhanced tropical skew and kurtosis, mainly restricted to the Atlantic and continental areas (Fig. S26). During the Deglaciation, the LGMR shows a pattern of negative skew in the North Atlantic consistent with most simulations with any kind of meltwater forcing. It further exhibits state-dependency in the spatial patterns of skewness and kurtosis and an increasing global mean kurtosis from LGM to Holocene (Fig. S5i). On the other hand, the LGMR overall shows very few areas of significant deviations from normality for LGM and Deglaciation and patterns that are generally more smooth than almost all simulations. However, the higher order moments are affected by the spatial averaging inherent in field reconstructions from individual proxy sites (c.f. Director and Bornn, 2015; McKinnon et al., 2016; Haylock et al., 2008). As such, a comparison at individual proxy locations could shed more light on model-data differences and similarities, but it is out





of the scope of this manuscript. As it stands, it remains unclear whether the LGMR has too little variability as indicated by its differences from both simulations and other proxy reconstructions.

### 5.2.3 Precipitation distributions show trend towards drier and less extreme years in the tropics from LGM to Holocene

For precipitation, mean changes are similarly varied as for surface temperature with a global mean LGM to Holocene wetting of $0.15 - 0.39 \frac{\text{mm}}{\text{d}}$. The ESMs and GCMs simulate some drying towards the Holocene over the tropical oceans, wetting occurs almost everywhere else and is particularly strong over Southeast Asia and the North American continent.

The moments are generally largest on annual timescales. In the tropics, ESMs and GCMs show some state-dependency, but not as much as for temperature. Tropical precipitation mostly decreases from the LGM to Holocene, especially over the ocean

and with it, the spread. During the Deglaciation, some high precipitation years remain, as indicated by the unchanged positive skewness and kurtosis. For the Holocene, however, all moments are generally smaller for annual precipitation in comparison to the LGM, indicating that the overall ranges of likely as well as extreme precipitation states have shrunk. Towards longer timescales, standard deviation decreases significantly and shows very little state-dependency. For skewness and kurtosis, the tropical peaks diminish, but instead areas of larger skew or kurtosis emerge in the mid- and high latitudes, especially for

the Deglaciation. So while annual distributions show that there are extreme precipitation years in the tropics, decadal and centennial distributions demonstrate that such extreme conditions rarely persist for whole decades or centuries. For skewness, dry regions are generally associated with positive skew since only a high value tail can exist for low mean precipitation. This is why precipitation mostly has positive skewness reflecting high precipitation extremes.

Some MPI-ESM runs and TraCE-21ka show bipolar skewness patterns between the hemispheres, but disagree on whether

the negative skewness is in the Northern (MPI-ESM) or Southern Hemisphere (TraCE-21ka). These bipolar patterns appear in response to meltwater forcing (c.f. Sec. 5.3). The spectra show little state-dependency on annual to multi-decadal timescales (Fig. 12i, j). For centennial timescales and longer, the Deglaciation shows a strong increase in variability, strongest in the tropics (Fig. S30 and S31), setting it apart from the LGM and Holocene. LGM and Holocene differ to a lesser degree, but start diverging on centennial timescales, although the simulations disagree whether LGM or Holocene has stronger spectral power.

### 815 5.3 Dependence of surface climate variability on external forcings

While ice sheet changes, meltwater fluxes and volcanism all have characteristic timescales, differences in these forcings cause variability changes on other timescales as well. Here, we compare simulations using ICE6G with GLAC1-D, with the latter having more extensive but lower Glacial ice cover in the Northern Hemisphere. The ice sheet reconstructions further have different temporal resolution, at $500$ and $100\,\text{yr}$ for ICE6G and GLAC1-D, respectively. While the reconstructions were inter-

polated for MPI-ESM r1–7, this difference in the underlying timescale will still affect centennial variability and likely explains some of the increased variability found for simulations using GLAC1-D in comparison to those using ICE6G (c.f. Fig. 6 and S27). However, the differences between the simulations using ICE6G and GLAC1-D are more complex. For the northern latitudes, the comparison indeed reveals a general association of GLAC1-D with larger standard deviation during the LGM and





Deglaciation (Fig. 6). On the other hand, GLAC1-D is associated with reduced standard deviation in parts of the Southern
Ocean during the Deglaciation and Holocene related to less variance in sea ice (Fig. S33 in the Supplement). These differ-
ences hold even for annual standard deviation (Fig. S15). Towards longer timescales, the simulations with GLAC1-D are both
more positively and negatively skewed. This holds in particular for the Deglaciation, for which this pattern appears already
on decadal scales. This is due to a relationship between skewness and the meltwater releases accompanying the respective ice
sheet reconstructions.

We find that a bipolar skewness pattern indicates a meltwater pulse in the area of negative skewness. GLAC1-D introduces
meltwater pulses, in particular MWP-1A and MWP-1B, mainly in the Northern Hemisphere. These lead to a freshening of
North Atlantic surface water and a slowdown of the Atlantic Meridional Overturning Circulation (AMOC). As a result, the
Northern Hemisphere cools and experiences more cold outliers (negative skewness), while the Southern Hemisphere warms. In
ICE6G, on the other hand, MWP-1B is a freshwater pulse mainly released into the Southern Ocean, counteracting and reducing
the dominating bipolar pattern. In TraCE-21ka, on the other hand, the most abrupt freshwater pulse (MWP-1A), is mostly
imposed as a freshwater flux into the Southern Ocean and as such leads to an opposite pattern in skewness. Kurtosis, similarly,
is stronger in GLAC1-D simulations during the LGM and Deglaciation, an effect that grows towards longer timescales. During
the Holocene temperature kurtosis is mostly smaller.

The moments of precipitation depend far less on the ice sheet reconstruction. Any effects are mostly confined to skewness,
where GLAC1-D is again linked to stronger skew, implying more extreme precipitation. The chosen ice sheet reconstruction
significantly impacts variability, especially for temperature, and behavior at the tails of the distributions across timescales.
However, there is considerable uncertainty in ice sheet reconstructions and corresponding meltwater releases (Stokes et al.,
2015; Abe-Ouchi et al., 2015; Ivanovic et al., 2016), an uncertainty that simulated variability thus retains. Many of the differ-
ences between simulations using GLAC1-D and ICE6G are the result of their associated meltwater protocols.

Locations of meltwater release are imprinted on the moments of the distribution, as outlined above. Further, there is a general
ranking of simulations by meltwater protocol with variability increasing from *no-melt* to *melt-uniform* and finally *melt-routed*.
Since simulations are generally believed to lack variability at least regionally (e.g. Laepple et al., 2023; Weitzel et al., 2024;
Rehfeld et al., 2018), this supports the usage of the *melt-routed* protocol. Locally, *melt-uniform* can result in larger moments
than *melt-routed*, while *melt-routed* generally shows large localized variability in connection to areas of meltwater injection.
Here, this is the North Atlantic for temperature, whereas for precipitation the equatorial Atlantic and Eastern equatorial Pacific
are most affected. The *no-melt* scenario is always linked to decreased variability.

Besides increased standard deviation, the *melt-routed* forcing also results in sustained negative temperature skew in the
North Atlantic described above during the Deglaciation, which is similar to the pattern found in the LGMR (Fig. 7 and S26).
Meltwater forcing also has the most significant impact on precipitation kurtosis among the forcings examined here. Since
meltwater forcing has such a strong association with variability, as well as the overall deglacial climate evolution (Snoll
et al., 2024), variability could help constrain meltwater releases. However, this would require identifying models with high
skill regarding simulated variability, currently hindered by a large uncertainties in reconstructed variability which could serve





as a benchmark of expected levels of surface climate variability. Furthermore, tuning meltwater to reproduce reconstructed variability alone is likely too simplified an approach (Weitzel et al., 2024)

In contrast to ice sheets and meltwater, volcanism has the largest impact on annual scales as is to be expected since volcanism introduces short-term cooling events. However, its impacts are evident on longer timescales as well, as is particularly apparent in the spectra, where volcanism raises the power across timescales (Fig. 12). For decadal to centennial timescales, the difference between the simulation with volcanic forcing (MPI-ESM r7) and those without is especially stark. Generally, volcanic forcing increases standard deviation, but with regional differences. It further reduces positive temperature skew across timescales and

kurtosis on annual scale as high temperature outliers become less and low temperature outliers more likely in response to the negative radiative forcing imposed. During the Holocene, volcanism counteracts otherwise hot centuries of the tropical ocean as indicated by the disappearance of strong positive skew and kurtosis in the tropical oceans present in MPI-ESM r1–6. Volcanism mainly affects the moments of temperature, while those of precipitation remain largely unchanged. To a lesser degree this also holds for ice sheet changes and meltwater forcing, such that the forcings leave an overall stronger imprint on temperature than

on precipitation.

## 5.4 Dependency of surface climate on model complexity suggest necessary minimal complexity

Our results suggest that there is a required minimal complexity for modeling the variability of surface climate. The EBM only reflects the linear response and represents mean changes far better than variability changes. It does not possess the complexity required to capture changes beyond the mean, which more dedicated simple models might (c.f. Lovejoy et al., 2021;

Schillinger et al., 2022). The increasing resemblance of the EBM's moments to those of the more complex models towards longer timescales (Fig. S5), on the other hand, suggests that at the global scale, centennial moments are dominated by the linear response to external forcings. This does not account for spatial patterns, though, which the EBM fails to capture.

The EMIC simulations provide a far better approximation of temperature variability, especially for standard deviation, but fall off for the extreme tails of the temperature distributions and for precipitation as a whole. The latter suggests that the EMICs

lack variability with respect to atmospheric dynamics. The spectral power of the EMICs is almost always on the lower end of the ensemble further indicating a lack of energy transfer between scales. As such, while they can match variability of proxy reconstructions on the global level (Zhu et al., 2019), they are limited for studies of regional variability. Since the EMICs included here have a more complex ocean than atmosphere model, using an EMIC more focused on atmospheric dynamics might improve results.

Beyond this, providing a ranking of simulated variability proves difficult due to the sparse spatial coverage of reconstructions as well as the lack of literature studying variability during the Deglaciation, although first attempts have been made (Weitzel et al., 2024). Moreover, at a certain level of complexity, here summarized as GCMs and ESMs, chosen forcings cause differences between simulations at least as much as the chosen model and its complexity. Due to substantial differences in forcing protocols, it can be hard to identify the source of differences between simulations from different models of similar complexity.

While there are some common patterns that emerge, the simulations also disagree in many areas with respect to magnitude and even direction of changes. This variety in the ensemble can be hidden when considering multi-model means. Since the differ-



ences are so large, no clear recommendation can be made as to a sufficient complexity for simulating variability. Moreover, due to large regional differences in model abilities, the best choice will depend on the scientific question to be answered.

## 6    Conclusions

The variability of surface climate has undergone considerable changes since the LGM along with an increase in GMST. The warming ranges from $3.0 - 6.6°C$ in the ensemble of fifteen transient simulations from models of varying complexity presented here. This result agrees with the estimates found in reconstructions, which also cover a large range. For the ensemble, we find that variability of surface temperature and precipitation depends on

- **timescale**: Towards longer timescales, the distributions of temperature show a decrease in variance, more skew (both
positive and negative corresponding to more temperature extremes) and an increase in kurtosis, i.e. enhanced non-Gaussianity. For precipitation, variance decreases, too, while changes to the higher order moments are mainly limited to the Deglaciation and differ between simulations and regions.

- **background state**: Overall, state-dependency of variability increases towards longer timescales. Beginning on decadal scales, the Deglaciation stands out as a period of enhanced variability in comparison to the LGM and Holocene. This
is marked by increased variance, more skew (both positive and negative) and larger kurtosis in most simulations. LGM and Holocene differ as well. The LGM has little overall temperature skewness and kurtosis, whereas there can be areas of large skewness and kurtosis during the Holocene. For precipitation during the LGM and Holocene, state-dependency can be found to some degree on all timescales and for all moments, but no clear patterns emerge.

- **forcing protocol**: Volcanism, meltwater forcing and ice sheet changes all affect the distributions of surface temperature
and precipitation beyond shifts in the mean. The effects of the forcings are strongest on their characteristic timescales, but can be detected on all timescales from annual to millennial. An ice sheet reconstruction with larger but lower ice sheets in the Northern Hemisphere increases temperature variance, absolute skewness and kurtosis there during the Deglaciation. For precipitation, larger ice sheet extents mainly introduce more skew on long timescales. The choice between uniform and routed meltwater injection also affects spatial variability on top of overall variability levels. Applying
no meltwater results in the smallest variability with the most homogeneous patterns, whereas simulations with routed meltwater generally have the largest overall and local variability. Locally, this manifests in raised variance, more skew (negative for temperature), and larger kurtosis. Since these patterns are associated with the place of meltwater injection, they affect temperature variability most in the North Atlantic as well as precipitation variability in the equatorial Atlantic and eastern equatorial Pacific. Volcanic forcing increases the variance and spectral power in simulations, in particular on
inter-annual to centennial scales. Beyond that, volcanic forcing changes the shape of distributions mainly for temperature as it decreases skewness — leading to shifts from positive to no or negative skew — and kurtosis values.

- **model complexity**: State-dependency becomes stronger for more complex models, which also show more complex patterns in time and space. In the EMICs investigated here, state-dependency is almost completely limited to temperature



variance, while skewness and kurtosis are close to zero for temperature and precipitation. The EBM sometimes exhibits
extreme spatial patterns, especially on short timescales, demonstrating its limitations in the simulation of variability.

To reach levels of variability comparable to reconstructions, simulations depend upon adequate levels of externally-forced variability, including from volcanic forcing, as well as internal variability. The contribution of internal variability requires at least the minimum complexity of the GCMs in this study. Nevertheless, comparison to some reconstructions of past climate suggests that simulations might lack variability, especially regionally and from multi-decadal timescales onward. While the
LGMR provided a first point of comparison, the comparison also raises questions about a potential loss of variability in field reconstruction methods when contrasted to other reconstructions. However, further conclusions necessitate an in-depth model-data comparison, which is limited by the small number of available proxy records, especially for precipitation. Since an improved understanding of the variability of surface climate is crucial, for example because of its relation to extremes and freshwater availability, a comparison to twentieth century observations could provide clearer evidence on the models' abilities
at least for short timescales. This could complement model evaluation efforts like CMIP and PMIP.

Several factors have emerged that could advance the evaluation of simulated variability and allow for a ranking of models: Improved benchmarks from reconstructions, more simulations with consistent protocols, ensemble of varying initial conditions, and high resolution snapshot simulations of past climates. The latter would allow an analysis of the changes in frequencies of extremes against regional mean changes and how extremes on short timescales transfer to longer timescales. As such, it could
elucidate the role of higher order moments in the evaluation of model simulations going forward.

Our results demonstrate that the Deglaciation stands out in comparison to the LGM and Holocene as a period of warming. However, an open question remains to what degree our results hold for future warming. To resolve this question, it is necessary to understand how much the increased variability is related to overall warming versus the initial state with large Northern Hemisphere ice sheets. Answering this question calls for a similar investigation for past warming periods as well as future
projections.

*Code and data availability.* The code and data to reproduce the analysis of this paper will be made available on zenodo upon publication and are now available upon request to EZ.

*Author contributions.* EZ and KR conceptualized this study and developed the methodology with input from NW. EZ implemented the analysis and visualized the results using simulations from MK, UM, LG, RI, PV and CW. EZ wrote the manuscript with input from KR, NW,
JB, MK and RI. All authors reviewed and discussed the manuscript.

*Competing interests.* The authors declare that they have no conflict of interest.





*Acknowledgements.* We thank Thomas Kleinen for discussion of the analysis and his simulation (MPI-ESM ch4). Further, we thank Moritz Adam for his feedback on a version of this manuscript.

EZ and NW have been supported by the Deutsche Forschungsgemeinschaft (DFG, German Research Foundation), project no. 395588486. This work has also been supported by the German Federal Ministry of Education and Research (BMBF), a Research for Sustainability initiative (FONA, www.fona.de) through the PalMod project, sub-projects 01LP1926C (JB and NW) and 01LP1917B (MK) during its second phase and sub-projects 01LP2310A (EZ), 01LP2311C (JB) and 01LP2302A (MK) during its third phase. KR is a member of the Machine Learning Cluster of Excellence, funded by the Deutsche Forschungsgemeinschaft (DFG, German Research Foundation) under Germany's Excellence Strategy – EXC number 2064/1 – Project number 390727645.




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
