# Peer review of "Patterns of changing surface climate variability from the Last Glacial Maximum to present in transient model simulations"

_EGUsphere, 2024_

## Referee Comment (RC2)

Review of

**Patterns of changing surface climate variability from the Last Glacial Maximum to present in transient model simulations**

by Ziegler et al.

**Recommendation**: *Accept with major revisions*

Summary The manuscript presents a detailed comparison of transient model simulations since the last Ice Age, using a hierarchy of physics-based 2D or 3D models, from simple energy balance models to Earth system models. The work is rigorous, well-motivated and well-written, and I am generally supportive of publication once a few key issues have been addressed.

**1 Scientific Comments**

**1.1 Death by moments**

I appreciate the authors' effort in setting up a general framework by which to analyze this ensemble of simulations. However, having to compare 4 moments across 15 simulations makes for a very large volume of information, bordering on overwhelming (witness the 41 figures in the supplement). This would be warranted if there was significant dynamical/physical insight to be gained from this assessment, but I found the paper lacking in this regard. For instance, what do changes in kurtosis (of temperature or precipitation) tell us about? Can this be tied to particular dynamics (e.g. convective clouds vs stratiform clouds produce precipitation distributions that can be distinguished by these moments). If not, is this really useful? In many places the insistence on painstakingly documenting the minute details of all four moments in these models makes for very bulky prose, from which this reader derived very few insights (e.g. L543-555). A statement like "The chosen ice sheet reconstruction has a limited impact on temperature kurtosis on all timescales analyzed here" does not help explain any dynamics. It also seems (e.g. L503) that it is broadly insensitive to many factors, so perhaps it is not a very useful indicator of anything? If so, why fill the paper with it?

I recommend using higher-order moments only when they can be connected to identifiable processes, and/or if they can be constrained by paleoclimate observations (proxies). Otherwise, this reads like a gratuitous exercise that multiplies figures without simultaneously enriching the content.

**1.2 Chapter or Paper?**

The methods' description, as well as the background, read more like a thesis chapter than a paper in a specialized journal, where some common understanding exists. Recalling the mathematical definition of the various moments (section 3.1), or of spectral quantities (section 3.2), seems overkill here. Either readers know it, or they can look up a standard stats textbook. Similarly, the opening

paragraph of section 3 is appropriate for a thesis chapter, but superfluous in a paleoclimate journal, where everyone either already understands this mathematical framework, or doesn't care enough about it for this exposition to matter. I recommend stripping down this exposition to the bare minimum, particularly if the code will be made available. The authors only need describe what they did in very general terms, and interested readers can go look at the code if they want to reproduce any of the results.

**2 Editorial Comments**

**L25** Insert full stop before "rather" and start a new sentence there.

**L114** "has been shown to reduce seasonal to interannual standard deviation" missing "the" before "seasonal"

**149** "normality assumptions might break down" -> this runs counter to my experience, whereby on long timescales, any averaging process (which is common in climate) makes things more Gaussian, by virtue of the central limit theorem. Can the authors explain why they expect normality to break down here?

**L173-174** : "They vary regarding simulation setup, applied forcings and model complexity." This is redundant with the previous paragraph.

**L189** "present-day": please define. The present is a singularly ambiguous notion in paleoclimatology [*Wolff*, 2007].

**L219** "2 kyr AP". Presumably "AP" stands for "after present", but since the present was not defined, this does not help. Also, this is the first time I see this acronym, so the authors should add a footnote to explain what they mean.

**Fig 2, caption** "with respect to the past 2 kyr": Is this the reference for everything? If so, please state explicitly in the text, upstream of this caption, so there is no ambiguity.

**Fig 3a** this is a very creative and helpful way to represent model complexity and help compare the various models used here. Most of the dimensions are qualitative, but atmospheric and oceanic resolution are two dimensions where one could be quantitative. Can the authors at least give a sense of the range of resolution spanned by the model ensemble?

**L296** "and EBM" : suggest "and *the* EBM", since there is only one here.

**L312** "from the end of the LGM to PI". Is PI the same as "the present"? See comments above, and please be consistent in what baseline is used for the modern era.

**L317-321** superflous paragraph

**L325** : "we remove the trend from the timeseries using a Gaussian filter" -> presumably this is a high-pass filter? Note that for discrete data a binomial filter is more justified.

**Sections 3.1, 3.2** tighten up so they read more like a paper and less like a dissertation chapter.

**L398** "a frequency range of $[2t_s,1000]$" -> technically, this is a range of periods

**L411** "Overall, MPI-ESM r1–r7 exhibit the largest temperature difference" Can this be tied to the equilibrium climate sensitivity of the various models?

L422]: "standard deviation provides a measure that increases with the spread of the distribution" -> technically, is IS a measure of the spread of a distribution. I find this wording needlessly mathematical.

**L437** : "On centennial scales, this lack of skewness agrees with the results for the LGMR." It should be pointed out that LGMR uses a Kalman Filter that assumes Gaussian state vectors, so no skewness could be reconstructed, even if it was there.

**Fig 5** : would it make sense for the $y$ axes to have units?

**L465** "standard deviation changes only locally" -> What does this mean? There are definitely some continental or basin scale patterns in some of the panels.

**L485** "differences in parametrization" influence the skewness. Which parameterizations might be responsible here? It may not be easy to guess, but if the authors have candidates in mind, they would be helpful to lay out here.

**L540; L554** : the word "significant" is used on those 2 lines. Do the authors mean "statistically significant" (if so, by which test?) or do they mean something like "substantial"? Please clarify.

**Fig 12** is very dense, and too small to distinguish many of the curves. The 6 PSD panels on the left have a curve/envelope in light gray, but there is no corresponding entry in the legend.

**L588** "we find temperature spectra that increase towards longer timescales" -> this certainly appears to be the case. Can the authors be more quantitative here? What scaling exponents are involved? How does it compare to observationally-derived exponents? Is there evidence of multifractality? If so, are the regime transitions occurring at the right timescales?

**L594** "relates to the simulated El Niño-Southern Oscillation". This seems to imply that those simulations exhibit enhanced ENSO activity with LGM boundary conditions. Is there any published explanation for this?

**L602-603** : again, this can be quantified with scaling exponents.

**L625** : "into the tropic" -> into the tropics

**section 4.6.4** : LGMR provides an ensemble of Kalman Filter samples. Is this taken into account here, or are the authors only considering the ensemble median?

**L672/3** : "Simulations that differ only by ice sheet reconstruction diverge most on long timescales, although differences can be found even for annual variability". This suggests that those boundary conditions affect the entire climate continuum, which is profound and deserves some commentary.

**Fig 13** : it took me a minute to figure out that the two gray lines correspond to PMIP end-members. The authors might want to clarify this in the figure caption.

**L758-759** : "Further, as a reanalysis product, the LGMR uses model simulations as priors and thus might be affected by a lack of variability in models." This intuition is incorrect. In the offline data assimilation flavor of ensemble Kalman FIlters used in products like LGMR or LMR, all temporal variability comes from the proxies; the model priors are used only to link

variables across space, or to link one variable to another (e.g. surface temperature to sea-level pressure). So this comment would only be true if it applies to spatial variability.

**last sentence** reads very generic. Please make more substantive, or dispense with it.

**References**

Wolff, E. W. (2007), When is the "present"?, *Quaternary Science Reviews*, *26*(25), 3023–3024, doi: 10.1016/j.quascirev.2007.10.008.

---

## Community Comment (CC1)

**Reply to Reviewer 1**

We thank the referee for the detailed and constructive criticism. In the following, we address the comments on a point-to-point basis. We also indicate the changes that we anticipate will be implemented in our manuscript. In summary, in response to the reviews 1 and 2 we will

- introduce our goals as key hypotheses to frame the manuscript

- reword individual sentences and shorten text in Methods and Results.

- remove Fig. 11 from the manuscript, include melt-water forcing in an additional panel in Fig. 1, rearrange Fig. 12 and modify labels in Figs. 2 and 3.

This will allow us to improve the manuscript substantially, and increase its impact.

**Response to scientific comments**

**Referee:** "The manuscript presents an analysis of global climate variability during the last deglaciation based on multi-model and reanalysis data. The authors collected climate model simulations of the last deglaciation with different climate models of different levels of complexities or experimental protocols. The authors analyzed global temperature and precipitation variabilities using multiple indicators for variabilities. The authors found increased climate variabilities during the last deglaciation than the LGM or Holocene with specific timescales and regions. The authors also find that the variability during the last deglaciation is affected by the complexity of the climate models or experimental design protocol.

I think this study's topic is well-suited for Climate of the Past, and the method and analysis of this study, particularly for introducing multiple variability indicators and analyzing both temperature and precipitation, is unique."

**Authors:** *We thank the reviewer for the thorough assessment of our study and appreciate that she/he noted that the study fills a gap in the literature.*

**Anticipated changes:** None.

**Referee:** "However, the manuscript needs additional work to improve the readability particularly for the following two points. Firstly, many figure panels, including supplemental figures (S41), are referenced in the manuscript (but some supplemental figures are not referenced), making it hard to follow. A multi-model study with global analysis may need many figure panels, but I had tough time understanding figures (Do Figs 6-11 need 24 or 27 panels?). I wonder if there is a better way to show figures in a more structured way to help readers."

**Authors:** *We agree with the reviewer that Figures 6-11 present a wealth of information, highlighting the differential role of model type, version, boundary conditions, and forcing on climate variability in transient simulations. In Fig. 6, for example, we investigated regional effects of meltwater and volcanic forcing on centennial-scale variability. Here, we were able to compute difference fields between different protocols, and could reduce the number of panels. Standard deviation patterns, for temperature, were sufficiently similar (and previously described in the literature, e.g. Rehfeld et al., ESD 2020), that we condensed these onto a latitudinal view in Fig. 6. However, to allow for intercomparison between Figs. 7-11, so skewness and kurtosis of temperature (7,8), as well as standard deviation and skewness of precipitation (9,10) we would prefer to keep the same Figure layout for a better comparison. Centennial-scale kurtosis of precipitation appears largely model-insensitive, and we therefore will remove Fig. 11. To enhance readability we will combine Fig. 25 (interannual kurtosis) and Fig. S24 (decadal kurtosis) and Fig. 11 (centennial kurtosis) for MPI-ESM r7 into one 3x3 plot, and refer to this in the discussion instead.*

**Anticipated changes:** We will remove Fig. 11 from the main manuscript (kurtosis of precipitation fields) and adjust the discussion.

**Referee:** "Secondly, the introduction section seems to lack information on what has been done regarding climate variability during the last deglaciation and what the knowledge gap is. As in the discussion section of this manuscript, there's a proxy study (e.g. Rehfeld et al. 2018) and climate modelling study (e. g. Zhu et al. 2019; Shi et al. 2022) on climate variability during the LGM or the last deglaciation. I think their methodology and results can be summarized in the introduction, and the authors can clarify what knowledge is lacking and what this study's strengths are. I also think stating a hypothesis in the introduction will help clarify the key points of this study."

**Authors:** *We thank the reviewer for this suggestion, which we agree would better frame the study, from the introduction to the discussion. We may hypothesize that (1) patterns of surface climate variability are state-dependent for the quasi-equilibrium conditions of the LGM and the Holocene, (2) the deglaciation, with the Earth system transitioning between a cold and a warm(er) climate state, stands out in the higher moments of precipitation and temperature distributions, (3) state-, model- and forcing-induced changes in variability are amplified with increasing timescales.*

**Anticipated changes:** We will include a paragraph on the current knowledge gap on LGM and glacial variability in the introduction, and reframe the last part of the introduction with the key hypotheses.

**Specific Comments:**

**Referee:** "L8-L9: The phrase "largely unexplored" might be too general. This sentence can be more specific based on previous knowledge gap or strength of this study."

**Authors:** *We note that, indeed, this sentence is very general, and a clear framing of (i) what we denote as climate variability and (ii) what the key knowledge gaps are improves the impact of the study.*

**Anticipated changes:** We will rewrite lines 8-9 of the abstract to highlight the focused definition of climate variability and the knowledge gap.

**Referee:** "L27-L28: I'm not sure what is unclear. Do you mean it is unclear whether LGMR (Osman et al. 2021) simulates accurate spatial patterns of climate variability?"

**Authors:** *The LGM reanalysis by Osman et al. draws on model simulations and a proxy dataset to reconstruct a spatio-temporal evolution of surface temperature changing from the LGM to the Holocene. Indeed, the LGMR reconstruction shows similar climate variability in the global mean to most models. However, by definition, the data assimilation procedure merges properties of the proxy data and the underlying model ensemble, following an algorithm. It is therefore not quite clear which one of these three aspects dominates in a reconstruction for a given region and timescale. However, patterns of centennial temperature variability change, in particular over the deglaciation, are smoother, and closer to normally distributed in LGMR than they are for most of the model simulations. Still, we cannot at this point clarify whether these patterns are accurate or not. Research into palaeoclimate data assimilation must clarify the impact of reconstruction methods on the higher moments of temperature, and ideally also precipitation, distributions.*

**Anticipated changes:** We will rephrase the statement in L27-28 as "A reanalysis of the LGM exhibits similar global mean variability to most of the ensemble. However, paleoclimate data assimilation combines model and proxy data information using a Kalman filter-based algorithm. More research is needed to disentangle their relative impact on reconstructed levels of variability."

**Referee:** "L72-L79: I'm not sure what the point of this paragraph is. I wonder if L71 and L80

can be directly connected to state the importance of climate variability and what proxy says on climate variabilities in the last deglaciation."

**Authors:** *We understand that the reviewer would prefer us to introduce climate variability more concisely and pragmatically with the deglaciation and proxy data in mind first. We agree that this would likely increase readability, and will re-order the paragraph so that it better leads to the abovementioned key hypotheses.*

**Anticipated changes:** We will reframe the paragraph in the introduction leading to the key hypotheses.

**Referee:** "L134-L140: I understand that one strength of skewness is that it can be an indicator of abrupt climate change, according to this paragraph. There would be a discussion paragraph on whether skewness in the deglaciation simulations can be an indicator of abrupt climate changes. "

**Authors:** *Indeed, we find clearly outstanding patterns of skewness and kurtosis change over the deglaciation in the transient simulations. Our simulation ensemble undergoes large-scale, sometimes abrupt, changes due to prescribed boundary conditions and forcing. Regionally there may be abrupt change due to internal dynamics of atmosphere, sea-ice, ocean or land surface. In light of this, focused future work could investigate this by drawing on joint analyses of high-resolution proxy data for key variables of tipping elements, with coupled model simulations.*

**Anticipated changes:** We will include this consideration in the introduction and discussion.

**Referee:** "L141-L151: As far as I understand, applying skewness and kurtosis to paleoclimate is new in this study, which can be emphasized."

**Authors:** *Indeed, we are not aware of other studies that, systematically or otherwise, investigated higher moments than standard deviation for paleoclimate.*

**Anticipated changes:** We will include a statement to the effect that we are, to our knowledge, the first to investigate higher-order moments in paleoclimate in the abstract and conclusion.

**Referee:** "L181: Is dd/m always used as GMP, global mean precipitation? Please clarify."

**Authors:** *Units are always in Celsius (for temperature) or mm/d for precipitation. In the text this is always noted. It is also the case for all analyses and figures.*

**Anticipated changes:** We will add a sentence clarifying that we always use Celsius/$\frac{mm}{d}$ as units for temperature/precipitation.

**Referee:** "L187: I don't understand what is different between MPI-ESM r1&r6 and r2&r5, as all columns in Table 1 are the same. Are they from simulations with different model parameters in Kapsch et al. (2022)? One way is to add a reference column in Table 1."

**Authors:** *This is correct. These simulations are described in Kapsch et al., (2022) and they differ in parameter choice.*

**Anticipated changes:** As suggested, we will clarify the differences between the simulations in the text by adding one sentence in the model section, and we will include a reference column in Table 1. For the description, we will change L189-191 to: "They use different sets ice sheet reconstructions – GLAC1-D or ICE-6G_C (in the following ICE6G, Peltier et al., 2015) – and vary by meltwater scenario. Further, a parameter for cloud formation was changed in r5–r7 to remove a cold bias found in r1–r4 (as detailed in the supporting information of Kapsch et al. 2022)."

**Referee:** "L465 & L470 "centennial" instead of "decadal and centennial"? Because Figure 6 say centennial standard deviation."

**Authors:** *We thank the reviewer for pointing this out. This should read "centennial".*

**Anticipated changes:** Correction to "centennial timescales".

**Referee:** "L513-L523: Based on Figures 7, 8 it is discussed that volcanic forcing impacts skewness and kurtosis during Holocene based on MPI-ESM r6 and r7 simulations. However, Figures 7, 8 and S12 make me feel that with the volcanic forcing, MPI-ESM simulations resemble HadCM3 simulations despite HadCM3 not having volcanic forcing. Are there any discussions for this model difference? "

**Authors:** *We indeed discuss in Sect. 4.3.3 the difference that volcanic forcing on the simulated higher-order moments of the surface climate distributions makes. Surprisingly, perhaps, the patterns generated by HadCM3 intrinsically (i.e., without volcanic forcing) are similar to those simulated by MPI-ESM with volcanic forcing. We have not spelled this out previously.*

**Anticipated changes:** We will explicitly mention the similarity of MPI-ESM r7 to the HadCM3 simulations in Sect. 4.3.3.

**Referee:** "L614-L618: Is it because (a) volcanic forcing or inter-annual to centennial variability, or (b) volcanic forcing does not correspond to timescale variation, but it can induce inter-annual to centennial variability?"

**Authors:** *Volcanic forcing both acts to stimulate temperature variability linearly (i.e., on the timescales on which it is active), as well as on timescales that are longer. This was noted, amongst others, in Ellerhoff et al., 2022, and we do as well, in the discussion, albeit without reference to the literature and without distinguishing the linear and nonlinear impact.*

**Anticipated changes:** We will rephrase the inital statement to "Volcanic forcing strongly impacts the spectrum of simulated surface temperatures. The MPI-ESM r7 run, which includes it, has the largest PSD [...]". We will further add the appropriate reference in the discussion.

**Referee:** "L694-L714 and Figure 13: I couldn't understand the point of this subsection, and why Figure 13 is necessary for discussing variability uncertainty. Please add some introduction."

**Authors:** *This section, and Fig. 13, are geared to give context for the expected mean-state change between the LGM and the Holocene. This is important, because one of our key goals is to understand whether the patterns in higher-order moments we see are state-dependent, i.e. dependent on the global-mean change. This section thus provides a backdrop to the discussion of variability change. We will revise the wording of this section to align with the hypotheses outlined in the introduction to improve readability.*

**Anticipated changes:** Rewording of title and introductory text of Sect. 5.1.

**Referee:** "L735-737: You mean that the meridional temperature gradient is enhanced during LGM as in Shi et al. (2022), but the variance is not increased like Shi et al. (2022)? If so, isn't it a significant result worth emphasizing and discussing further?"

**Authors:** *Changes in the meridional surface temperature gradient have been suggested to drive interannual temperature variability change between the LGM and the Holocene on multi-centennial to millennial temperature variability (Rehfeld et al. 2018). Furthermore, Shi et al., 2022 suggested that, for the extratropics, there is a significant spatial correlation between the gradient change, and the variability pattern that they link to interannual variability. They also, indeed, suggest that temperature variability is increased by some 20% compared to the Holocene based on a PMIP3/PMIP3 model ensemble, which is consistent with the 25-31% increase found in the PMIP3 ensemble considered in Rehfeld et al., (2018). In our ensemble we found that, for some models and some regions, this notion holds, whereas for other regions and models there is no correlation between the temperature gradient and variability change. Overall we do find an enhanced temperature variability, and we do find an enhanced temperature gradient in the LGM, which is consistent with the previous*

*studies. We will emphasize this in the revision.*

**Anticipated changes:** Revision of text in Sect.5.2.1 to expand on and better present the results with respect to the relationship between the change in temperature gradient, and the change in surface temperature variability.

**Referee:** "L744-L745: In addition to long-term memory, there's transient forcing during 23 to 19 ka (Ivanovic et al. 2016), unlike equilibrium LGM simulation at 21ka."

**Authors:** *We thank the referee for pointing out that the transient change in forcing over the deglaciation is another external source of variability on centennial to millennial timescales. We missed to include it in this statement because the previous paragraph was referring to PMIP3/PMIP4 simulations, where these do not play a role. For the revision we will more clearly distinguish the two types of simulations in this section.*

**Anticipated changes:** We will reword the sentence to read "Since the differences are especially apparent on longer timescales, this might point towards long-term memory effects or transient forcing missing in such equilibrium simulations.".

**Referee:** "L884 (minor): Why "using an EMIC more focused on atmospheric dynamics" , unlike [using a GCM when focusing on climate variability] "

**Authors:** *Indeed, dependent on the timescale of key interest a GCM may be computationally efficient enough to be suitable for simulation of centennial to millennial climate variability (as evidenced by the selection of simulations we draw on in our study). We meant to emphasize the fact that most EMICs have a reduced atmospheric complexity, which limits their simulated variablity (e.g., Schillinger, Ellerhoff et al., 2022).*

**Anticipated changes:** We will rephrase the sentence to "The EMICs included here have reduced atmospheric complexity. This will affect simulated variability and could be different in other EMICs as it is for GCMs."

**Referee:** "L892-L893 (minor): Each simulation from previous articles used in this study focused specifically on the atmosphere or ocean processes of the last deglaciation, which is one primary reason the model complexity or experimental design differs. Even so, it's a great opportunity to discuss good choices on the scientific question of climate variability."

**Authors:** *We appreciate this suggestion to highlight a) reasons for the diversity of simulations and models and b) make a suggestion for an experimental design geared towards climate variability.*

**Anticipated changes:** We will add "Due to substantial differences in forcing [and boundary condition] protocols [inherently arising from different research foci], it can be hard to identify the source of difference between simulations from different models of similar complexity" in l888-889. Furthermore, we will rephrase in l892-893 to "An experimental design geared towards understanding the roles of feedbacks on surface climate variability must take into account external forcing and boundary condition changes, distinguishing interactive effects and prescribed changes in boundary conditions which may, or may not, be physically consistent with the climate evolution. Given the impact of meltwater forcing and its uncertainties, simulations with interactive ice sheets are of particular interest to the study of climate extremes in response to mean changes."

**Referee:** "L895-L925 (minor): The sentences overlap with the first paragraph of the discussion section. Please consider describing brief conclusions. (or merged with the discussion section? )"

**Authors:** *We thank the reviewer for this suggestion. We will revise the first sentences in the Conclusions focusing on closing the hypothesis-bracket from the introduction. This should reduce the overlap.*

**Anticipated changes:** We will rephrase the first two sentences in the conclusions.

**Referee:** "Figure 1a: EPICA Dome C (Jouzel et al. 2007) and NGRIP (Andersen et al. 2004) presents local temperature change at the ice core site, so it looks strange the vertical axis is represented as GMST. Please clarify the vertical axis."

**Authors:** *We thank the reviewer for the attention to detail. For easier visualization of both, Antarctic and Greenland temperature change, ice-core records were scaled to GMST assuming a polar amplification of 2. We will include the details of the calibration and scaling in the Figure caption.*

**Anticipated changes:** We will add the relevant calibration detail in the caption of Fig. 1.

**Referee:** "Figure 1c: this panel presents sea-level change, but it would fit better including meltwater input as the timeseries of meltwater. While meltwater input would differ between models (e.g. Snoll et al., 2024), it provides an essential information as the meltwater is discussed as the critical factor in climate variabilities. "

**Authors:** *We thank the reviewer for this suggestion. Meltwater input should, in principle, be related to the time-derivative of the sea-level curve and ice-sheet extent. We will include Meltwater forcing timeseries calculated from ICE-6G and GLAC1-D in the plot to provide the reference for the climate variability discussion.*

**Anticipated changes:** We will add a panel in Fig. 1 with meltwater forcing.

**Referee:** "Figure 12: What does "PMIP3" mean? Does it come from Li et al. (2013)? Please clarify in the caption and results section."

**Authors:** *We thank the referee for the careful reading. In Fig. 12 we denote by PMIP3 the interannual to decadal scale change in climate variability computed in Rehfeld et al., 2018 based on a PMIP3 simulation ensemble. The bar denoted 'reconstructed' refers to the multicentennial to millennial-scale change in variability reconstructed from (mostly) marine palaeoclimate proxy datasets.*

**Anticipated changes:** We will adapt the last sentence of the figure caption to read: "In panel d, Rehfeld et al. (2018)'s estimated range of the multi-centennial to millennial LGM-to-Holocene variance ratio based on proxy reconstructions (reconstructed) and interannual variability based on the PMIP3 ensemble (PMIP3) are marked for comparison."

**References**

Ellerhoff, B., Kirschner, M. J., Ziegler, E., Holloway, M. D., Sime, L., and Rehfeld, K.: Contrasting State-Dependent Effects of Natural Forcing on Global and Local Climate Variability, Geophysical Research Letters, 49, e2022GL098 335, `https://doi.org/10.1029/2022GL098335`, 2022.

Rehfeld, K., Hébert, R., Lora, J. M., Lofverstrom, M., and Brierley, C. M.: Variability of Surface Climate in Simulations of Past and Future, Earth System Dynamics, 11, 447–468, `https://doi.org/10.5194/esd-11-447-2020`, 2020.

---

## Community Comment (CC2)

[Figure]

**Figure 6.** Comparing tephra and ice-core-based VSSI time series over the Holocene. (a) VSSI time series derived from tephra records. (b) HolVol ice-core-based VSSI time series (Sigl et al., 2022). Gray shading shows the period used to base eruption statistics for the synthetic time series. (from Schindlbeck-Belo 2024)

*"We emphasize that users wanting the most accurate reconstruction of VSSI over the last glacial cycle could consider using a merged product, for example by concatenating the HolVol ice core time series with PalVol for the period, which occurs before the beginning of HolVol."* (from Schindlbeck-Belo 2024)

[Figure]

**Figure 1.** Climate responses and external forcing during the past 26k years: (a) global mean temperature anomaly (w.r.t. 1960-1989) as simulated by MPI-ESM, captured in ice cores from Antarctica (EPICA Dome C, Jouzel et al., 2007) and Greenland (NGRIP, Andersen et al., 2004) and reconstructed in the LGM reanalysis (Osman et al., 2021), (b) global mean precipitation as simulated by the Earth System Model MPI-ESM, (c) sea level change (Grant et al., 2012) (d) atmospheric $CO_2$ (Köhler et al., 2017) and (e) $CH_4$ levels (Köhler et al., 2017), (f) daily insolation at 65°N and 65°S at the summer solstice (Huybers and Eisenman, 2006), (g) solar constant from one ensemble member generated as surrogate data based on Steinhilber et al. (2009) following Ellerhoff and Rehfeld (2021) (comparison with Steinhilber et al. (2009) in Fig. S2 in the Supplement), and (h) volcanic forcing TephraSynthIce (Schindlbeck-Belo et al., 2023).

---

## Author Comment (AC1)

**Reply to reviewer 2**

We thank the referee for the detailed and constructive criticism and the appreciation shown for our work. In the following, we address the comments on a point-by-point basis. We also indicate the changes we will implement in our manuscript upon revision. In summary, in response to the reviews 1 and 2 we will

- introduce our goals as key hypotheses to frame the manuscript

- reword individual sentences and shorten text in Methods and Results.

- remove Fig. 11 from the manuscript, include melt-water forcing in an additional panel in Fig. 1, rearrange Fig. 12 and modify labels in Figs. 2 and 3.

We are confident this will improve the scientific relevance, impact and readability of the study.

**Response to scientific comments**

**Referee:** "The manuscript presents a detailed comparison of transient model simulations since the last Ice Age, using a hierarchy of physics-based 2D or 3D models, from simple energy balance models to Earth system models. The work is rigorous, well-motivated and well-written, and I am generally supportive of publication once a few key issues have been addressed."
**Reply:** *We thank the referee for this positive assessment of our work and address the issues raised in the following.*
**Anticipated changes:** None.

**Referee:** "1.1 Death by moments
I appreciate the authors' effort in setting up a general framework by which to analyze this ensemble of simulations. However, having to compare 4 moments across 15 simulations makes for a very large volume of information, bordering on overwhelming (witness the 41 figures in the supplement). This would be warranted if there was significant dynamical/physical insight to be gained from this assessment, but I found the paper lacking in this regard. For instance, what do changes in kurtosis (of temperature or precipitation) tell us about? Can this be tied to particular dynamics (e.g. convective clouds vs stratiform clouds produce precipitation distributions that can be distinguished by these moments). If not, is this really useful? In many places the insistence on painstakingly documenting the minute details of all four moments in these models makes for very bulky prose, from which this reader derived very few insights (e.g. L543-555). A statement like "The chosen ice sheet reconstruction has a limited impact on temperature kurtosis on all timescales analyzed here" does not help explain any dynamics. It also seems (e.g. L503) that it is broadly insensitive to many factors, so perhaps it is not a very useful indicator of anything? If so, why fill the paper with it?
I recommend using higher-order moments only when they can be connected to identifiable processes, and/or if they can be constrained by paleoclimate observations (proxies). Otherwise, this reads like a gratuitous exercise that multiplies figures without simultaneously enriching the content."

**Reply:** *We thank the referee for the comment and share the concern about the volume of information provided. We attempted to separate significant from non-significant changes, providing the latter only in the supplement as a point of reference. We acknowledge that this separation could be more strictly implemented.Rigorously studying how distributions of temperature and precipitation change on the long-term has never been attempted. We therefore chose to provide comprehensive results of the analysis, even those that are inconclusive or show no difference with timescale, state or forcing. This is to provide a basis for future analyses, exploration and exploitation. As these*

*plots are only part of the supplement, we think keeping them is acceptable. For the main text, we will further narrow in on the relevant changes as suggested by the referee. To this end, we will move Fig. 11 (kurtosis of precipitation) into the supplement (and narrow in on the timescale-dependent changes by combining it with Fig. S24 and S25 and showing this for only one model). We will further tighten the text and reduce the level of detail provided.*

*Considering the heterogeneity of the results between models and simulations, a certain amount of information is necessary to highlight relevant processes and dynamics. On interannual to centennial scales, surface temperature and precipitation are the end members of a chain of dynamical processes in the climate system. It is difficult, at this point, to confidently establish links between processes and all aspects of surface climate variability. We do identify a range of processes that could link dynamics to the distributions of surface temperature and precipitation, but there is strong model diversity. We agree that pinpointing dynamics and feedbacks is of interest. Further clarification is hopefully possible when differences in model setups, boundary conditions and forcing could become a target for the next generation of model intercomparison projects.*

*A more in-depth comparison of simulated to reconstructed surface climate variability would be desirable. However the coverage in time and space of proxy records is limited, especially for precipitation-related variables. The LGM reanalysis is the only field reconstruction that covers the whole time period. Uncertainties in model-data comparison on variability are large and narrowing down the role of dynamical climate processes by proxy forward modeling, single-model-perturbed-parameter experiments and coordinated protocols is future work we will highlight as important in the revision.*

**Anticipated changes:** We will move Fig. 11 to the supplement and combine it with Fig. S24 and S25. We will remove detail in Sec. 4.3 – 4.5 to narrow in on the most notable features of the analysis.

**Referee:** "1.2 Chapter or Paper?
The methods' description, as well as the background, read more like a thesis chapter than a paper in a specialized journal, where some common understanding exists. Recalling the mathematical definition of the various moments (section 3.1), or of spectral quantities (section 3.2), seems overkill here. Either readers know it, or they can look up a standard stats textbook. Similarly, the opening paragraph of section 3 is appropriate for a thesis chapter, but superfluous in a paleoclimate journal, where everyone either already understands this mathematical framework, or doesn't care enough about it for this exposition to matter. I recommend stripping down this exposition to the bare minimum, particularly if the code will be made available. The authors only need describe what they did in very general terms, and interested readers can go look at the code if they want to reproduce any of the results"

**Reply:** *We thank the reviewer for the feedback and agree that the methods section is intentionally detailed. We observed that for those who are unfamiliar with higher-order moments this section is helpful, for those who are familiar it may not be necessary for an intuitive understanding for following the manuscript. We do think having some explanation readily available instead of relying on textbooks is beneficial. Still, we will condense the section, and move parts of it to the supplement. Sections 3.1 and 3.2 will then focus on the details vital to understanding the results.*

**Anticipated changes:** We will shorten sections 3.1 and 3.2 and move some details to the supplementary material.

**Response to editorial comments**

**Referee:** "L25 Insert full stop before 'rather' and start a new sentence there."
**Anticipated changes:** We will make this change as suggested.

**Referee:** "L114 "has been shown to reduce seasonal to interannual standard deviation" missing "the" before "seasonal""
**Anticipated changes:** *We will add the "the" as suggested.*

**Referee:** "L149 "normality assumptions might break down" → this runs counter to my experience, whereby on long timescales, any averaging process (which is common in climate) makes things more Gaussian, by virtue of the central limit theorem. Can the authors explain why they expect normality to break down here? "
**Reply:** In the stationary system considered in an equilibrium climate model simulation fluctuations will indeed, at some point, average out (Central Limit Theorem). In reality, the Earth system is not stationary, as forcings and boundary conditions change over time. As we state in lines 327/328 we assume weak stationarity of our signals after detrending. Abrupt shifts, land-surface and boundary condition changes may impact shorter than millennial timescales, and thus the shapes of the distributions, which become non-Gaussian.
**Anticipated changes:** We will clarify the statement to read "normality assumptions might break down under a non-stationary climate evolution".

**Referee:** "L173-174 : "They vary regarding simulation setup, applied forcings and model complexity." This is redundant with the previous paragraph."
**Anticipated changes:** We will edit this to remove the redundancy.

**Referee:** "L189 "present-day": please define. The present is a singularly ambiguous notion in paleoclimatology [Wolff, 2007]."
**Reply:** *To account for this ambiguity, we have defined present-day in the beginning of the introduction, on p. 2 footnote 1.*
**Anticipated changes:** None.

**Referee:** "L219 "2 kyr AP". Presumably "AP" stands for "after present", but since the present was not defined, this does not help. Also, this is the first time I see this acronym, so the authors should add a footnote to explain what they mean."
**Reply:** *AP was defined in the first footnote together with BP.*
**Anticipated changes:** None.

**Referee:** "Fig 2, caption "with respect to the past 2 kyr": Is this the reference for everything? If so, please state explicitly in the text, upstream of this caption, so there is no ambiguity"
**Reply:** *Whenever discussing mean changes in the models, we do indeed calculate anomalies with respect to the past 2 kyr. We will make this clear in the revised version of the manuscript.*
**Anticipated changes:** We will insert a statement about mean anomalies being calculated with respect to the past 2 kyr in the text and adapt the text in the figure.

**Referee:** "Fig 3a this is a very creative and helpful way to represent model complexity and help compare the various models used here. Most of the dimensions are qualitative, but atmospheric and oceanic resolution are two dimensions where one could be quantitative. Can the authors at least give a sense of the range of resolution spanned by the model ensemble?"
**Reply:** *We thank the referee for the feedback. Indeed, the dimensions regarding resolutions are*

*quantitative based on horizontal and vertical resolution as outlined in Sec. S2.2 of the supplement. We agree that highlighting this in the figure would be helpful. We will update the figure accordingly, while maintaining legibility.*
**Anticipated changes:** We will add numbers to the levels for resolution in Fig. 3 and labels for the common models resolution to place them in the hierarchy.

**Referee:** "L296 "and EBM" : suggest "and *the* EBM", since there is only one here"
**Anticipated changes:** We will make this change as suggested.

**Referee:** "L312 "from the end of the LGM to PI". Is PI the same as "the present"? See comments above, and please be consistent in what baseline is used for the modern era."
**Reply:** *We agree that a consistent terminology is important and will clarify the definition of PI used in Osman et al. (2021), which is 1000–1850 CE.*
**Anticipated changes:** We will change the sentence to read "The resulting reanalysis estimates a global warming of $7.0 \pm 1.0°$ C from the end of the LGM to PI (with PI defined as 1000–1850 CE), as it is contains a LGM state colder than reconstructed elsewhere (c.f. Annan et al., 2022; Tierney et al., 2020; Shakun and Carlson, 2010).".

**Referee:** "L317-321 superflous paragraph"
**Anticipated changes:** We will shorten the methods as discussed above.

**Referee:** "L325 : "we remove the trend from the timeseries using a Gaussian filter" → presumably this is a high-pass filter? Note that for discrete data a binomial filter is more justified."
**Reply:** *Indeed, we apply the Gaussian filter as a high-pass filter. We appreciate the notion of the binomial filter being useful for use with discrete data. For reasons of simplicity and the benign and conservative properties of the Gaussian filter, we apply it for the removal of the trends. As a Gaussian filter is also suitable for irregularly sampled timeseries, we hope that using it further simplifies future comparison to proxy data.*

**Referee:** "Sections 3.1, 3.2 tighten up so they read more like a paper and less like a dissertation chapter."
**Anticipated changes:** We will shorten the methods as discussed above when addressing the corresponding scientific comment.

**Referee:** "L398 "a frequency range of [2ts,1000]" → technically, this is a range of periods"
**Reply:** *We thank the referee for catching this and will change this upon revision.*
**Anticipated changes:** We will change "frequency range" to "period range".

**Referee:** "L411 "Overall, MPI-ESM r1–r7 exhibit the largest temperature difference" Can this be tied to the equilibrium climate sensitivity of the various models?"
**Reply:** *Unfortunately, the equilibrium climate sensitivity (ECS) is not documented for all simulations. The temperature difference depends on model tuning and ECS, but also boundary conditions (as indicated by the 1 degree difference between ICE6G and GLAC1D runs where the same model version was used). Thus we cannot infer ECS based on the LGM-to-Holocene temperature difference, although we agree that the role of climate sensitivities in fostering climate variability would be interesting to study.*
**Anticipated changes:** None.

**Referee:** "L422: "standard deviation provides a measure that increases with the spread of the

distribution" → technically, is IS a measure of the spread of a distribution. I find this wording needlessly mathematical."

**Anticipated changes:** We will change this sentence to "Standard deviation represents the spread of the distribution (c.f. Sect. 3.1).".

**Referee:** "L437 : "On centennial scales, this lack of skewness agrees with the results for the LGMR." It should be pointed out that LGMR uses a Kalman Filter that assumes Gaussian state vectors, so no skewness could be reconstructed, even if it was there."

**Reply:** *Indeed, a Kalman filter's assumption of normality will affect the results of the data assimilation (depending on the timescale of assimilation versus that of the final reconstruction). This effect is complicated by the impact of model prior and resolution of the individual proxy records. As such, our statistical analysis of the LGMR are to be interpreted with caution. The impact of data assimilation reconstruction methods on higher moments requires further research. We will clarify this in the text when discussing the LGMR in Sec. 5.2.2.*

**Anticipated changes:** We will add to the discussion of the LGMR in Sec. 5.2.2 and its comparison to highlight potential uncertainty due to potential impacts of the reconstruction method.

**Referee:** "Fig 5 : would it make sense for the y axes to have units"

**Reply:** *Since units are not common across rows or columns, it would be complicated to indicate them in the figure at is stands. Instead, we suggest adding the units to the figure caption.*

**Anticipated changes:** We will change the second sentence in the figure caption to "For all simulations standard deviation (left column, in units of °C for temperature and $\frac{mm}{d}$ for precipitation), skewness (middle column, dimensionless) and kurtosis (right column, dimensionless) are shown.".

**Referee:** "L465 "standard deviation changes only locally" -¿ What does this mean? There are definitely some continental or basin scale patterns in some of the panels."

**Reply:** *We agree that "locally" may not be the best word choice as it suggests the grid box level. We meant to indicate that across models most areas see only small changes with changing ice sheets (Fig. 6, panels a-f). We will change the sentence to better reflect the results and thank the referee for pointing out this imprecise wording.*

**Anticipated changes:** We will change the sentence to: "Changes in the spread of the distribution, as expressed in standard deviation, are regionally limited in response to the prescribed ice sheet reconstruction (Fig. 6 a–f)."

**Referee:** "L485 "differences in parametrization" influence the skewness. Which parameterizations might be responsible here? It may not be easy to guess, but if the authors have candidates in mind, they would be helpful to lay out here"

**Reply:** *Here, specifically the difference in parametrizations between the model versions of MPI-ESM relate to the formation of clouds. The changed parameter (csecfrl) describes the threshold between cloud water and ice according to the Wegener-Bergeron-Findeisen (WBF) process. This describes the preferred growth of ice crystals at the expense of large water droplets in supercooled and supersaturated conditions. The modification leads to warmer PI and LGM climates in r5 and r6 in comparison to r1 and r2. r1 and r2 were found to be too cold in comparison to reconstructions (for further details see supporting information of Kapsch et al. 2022). We will specify that the changed parametrization relates to cloud formation.*

**Anticipated changes:** We will change "This indicates that the differences in parametrization influence the skewness." to "This indicates that the difference in cloud formation parametrization influence the skewness.". We will further clarify the description of the simulations in L189-191 to say "They use different sets ice sheet reconstructions – GLAC1-D or ICE-6G_C (in the following

ICE6G, Peltier et al., 2015) – and vary by meltwater scenario. Further, a parameter for cloud formation was changed in r5–r7 to remove a cold bias found in r1–r4 (as detailed in the supporting information of Kapsch et al. 2022)."

**Referee:** "L540; L554 : the word "significant" is used on those 2 lines. Do the authors mean "statistically significant" (if so, by which test?) or do they mean something like "substantial"? Please clarify."
**Reply:** *This does indeed mean statistically significant. It refers to the results of the significance tests as described in Sec. 3.1, (t-tests testing for the hypothesis of normality). The results of the significance tests are summarized as the percentages of grid boxes with positive or negative significant changes below skewness and kurtosis maps.*
**Anticipated changes:** None.

**Referee:** "Fig 12 is very dense, and too small to distinguish many of the curves. The 6 PSD panels on the left have a curve/envelope in light gray, but there is no corresponding entry in the legend."
**Reply:** *The curves in grey correspond to the sensitivity set as described in the caption. We will remove them from the figure in the revised version of the manuscript. To enlarge the panels, we will further move both columns with the spectral ratios below the other panels.*
**Anticipated changes:** We will remove the sensitivity set from the figure and move panels d,e,i and j below the rest of the figure.

**Referee:** "L588 "we find temperature spectra that increase towards longer timescales" → this certainly appears to be the case. Can the authors be more quantitative here? What scaling exponents are involved? How does it compare to observationally-derived exponents? Is there evidence of multifractality? If so, are the regime transitions occurring at the right timescales?"
**Reply:** *Indeed, an analysis of the scaling by computing scaling factors and identifying scale breaks would be of great interest and could contribute to the debate on the nature of scaling, the existence and potential timescale of scale breaks and how these relate to glacial versus interglacial conditions (Huybers & Curry, 2006, Nilsen et al., 2016, Lovejoy, 2015, Rypdal et al., 2013). An in-depth analysis of simulated scaling, its dependence on mean state, region and forcing as well as a comparison the results to reconstructions (as e.g., in Ellerhoff & Rehfeld, 2021 for the last 2000 years) is, unfortunately, out of the scope here. We add this as a potential future target of investigation to the discussion.*
**Anticipated changes:** We will add to the discussion of scaling with respect to possible future analysis of scaling.

**Referee:** "L594 "relates to the simulated El Niño-Southern Oscillation". This seems to imply that those simulations exhibit enhanced ENSO activity with LGM boundary conditions. Is there any published explanation for this?"
**Reply:** *Indeed, the MPI-ESM simulations exhibit an enhanced ENSO signal during the LGM. However this has been noted elsewhere (e.g. Ellerhoff & Rehfeld 2021). None of the other simulations suggest a similar feature.*
**Anticipated changes:** We will note that the multi-model ensemble does not suggest higher ENSO activity.

**Referee:** "L602-603 : again, this can be quantified with scaling exponents."
**Reply:** *We agree that such an analysis is a promising direction for future work (see response to change suggested for L588*

**Anticipated changes:** See response to comment on L594.

**Referee:** "L625 : "into the tropic" → into the tropics"
**Anticipated changes:** We will change this as suggested.

**Referee:** "section 4.6.4 : LGMR provides an ensemble of Kalman Filter samples. Is this taken into account here, or are the authors only considering the ensemble median?"
**Reply:** *Indeed, we consider the ensemble mean. We will clarify this upon revision of the manuscript.*
**Anticipated changes:** We will add to Sec. 2.3, the caption of Fig. 5 and Sec. 4.6.4 to clarify that we use the ensemble mean for comparison.

**Referee:** "L672/3 : "Simulations that differ only by ice sheet reconstruction diverge most on long timescales, although differences can be found even for annual variability". This suggests that those boundary conditions affect the entire climate continuum, which is profound and deserves some commentary."
**Reply:** *We agree that this is an intriguing result, deserving further discussion. It clearly highlights that complex interactions of slow-changing boundary conditions can impact shorter timescales in the system. This was implicitly mentioned in Sect. 4.3 and 5.3 and the conclusions. We will highlight this upon revision of the manuscript.*
**Anticipated changes:** We will highlight that slow components can impact variability on faster timescales in Sect. 5.3 explicitly.

**Referee:** "Fig 13 : it took me a minute to figure out that the two gray lines correspond to PMIP end-members. The authors might want to clarify this in the figure caption."
**Reply:** *Indeed, the figure caption is incomplete and we thank the referee for catching this. We will change it in the revision.*
**Anticipated changes:** We will adapt the last sentence of the figure caption to read: "In panel d, Rehfeld et al. (2018)'s estimated range of the multi-centennial to millennial LGM-to-Holocene variance ratio based on proxy reconstructions (reconstructed) and interannual variability based on the PMIP3 ensemble (PMIP3) are marked for comparison."

**Referee:** "L758-759 : "Further, as a reanalysis product, the LGMR uses model simulations as priors and thus might be affected by a lack of variability in models." This intuition is incorrect. In the offline data assimilation flavor of ensemble Kalman FIlters used in products like LGMR or LMR, all temporal variability comes from the proxies; the model priors are used only to link variables across space, or to link one variable to another (e.g. surface temperature to sea-level pressure). So this comment would only be true if it applies to spatial variability"
**Reply:** *We thank the reviewer for this comment. We will adapt the discussion of the LGMR.*
**Anticipated changes:** We will change the sentence to "Further, as a reanalysis product, the LGMR uses model simulations as priors, which might affect the spatial patterns found.".

**Referee:** "last sentence reads very generic. Please make more substantive, or dispense with it."
**Anticipated changes:** We will replace the sentence: "Interactions between dynamics, forcings and mean state lead to complex changes in the distributions of surface climate variables. This implies potential changes to extremes on timescales from years to centuries, requiring further investigation."

**References**

Ellerhoff, B., and Rehfeld, K. (2021). Probing the timescale dependency of local and global variations in surface air temperature from climate simulations and reconstructions of the last millennia.

Physical Review E, 104(6), 1–14. `https://doi.org/10.1103/PhysRevE.104.064136`

Huybers, P. and Curry, W.: Links between Annual, Milankovitch and Continuum Temperature Variability, Nature, 441, 329–332, `https://doi.org/10.1038/nature04745`, 2006.

Kapsch, M. L., Mikolajewicz, U., Ziemen, F., and Schannwell, C.: Ocean Response in Transient Simulations of the Last Deglaciation Dominated by Underlying Ice-Sheet Reconstruction and Method of Meltwater Distribution, Geophysical Research Letters, 49, 1–11, `https://doi.org/10.1029/2021GL096767`, 2022.

Lovejoy, S. (2015). A voyage through scales, a missing quadrillion and why the climate is not what you expect. Climate Dynamics, 44(11–12), 3187–3210. `https://doi.org/10.1007/s00382-014-2324-0`

Nilsen, T., Rypdal, K., and Fredriksen, H. B.: Are There Multiple Scaling Regimes in Holocene Temperature Records?, Earth System Dynamics, 7, 419–439, `https://doi.org/10.5194/esd-7-419-2016`, 2016.

Osman, M. B., Tierney, J. E., Zhu, J., Tardif, R., Hakim, G. J., King, J., and Poulsen, C. J.: Globally Resolved Surface Temperatures since the Last Glacial Maximum, Nature, 599, 239–244, `https://doi.org/10.1038/s41586-021-03984-4`, 2021.

Rypdal, K., Østvand, L., Rypdal, M. (2013). Long-range memory in Earth's surface temperature on time scales from months to centuries. Journal of Geophysical Research: Atmospheres, 118(13), 7046–7062. `https://doi.org/10.1002/jgrd.50399`